# Enhanced pore space analysis by use of μ-CT, MIP, NMR, and SIP

Zeyu Zhang [1], Sabine Kruschwitz [2, 3], Andreas Weller [4], Matthias Halisch [5]

[1] Southwest Petroleum University, School of Geoscience and Technology, 610500 Chengdu, China

[2] Federal Institute for Material Research and Testing (BAM), D-12205 Berlin, Germany

[3] Technische Universität Berlin, Institute of Civil Engineering, D-13355 Berlin, Germany

[4] Clausthal University of Technology, Institute of Geophysics, D-38678 Clausthal-Zellerfeld, Germany

[5] Leibniz Institute for Applied Geophysics (LIAG), D-30655 Hannover, Germany

*Correspondence to*: Zeyu Zhang (zeyuzhangchina@163.com)

**Abstract**
We investigate the pore space of rock samples with respect to different petrophysical parameters using
various methods, which provide data upon pore size distributions, including micro computed tomography
(μ-CT), mercury intrusion porosimetry (MIP), nuclear magnetic resonance (NMR), and spectral induced
polarization (SIP). The resulting cumulative distributions of pore volume as a function of pore size are
compared. Considering that the methods differ with regard to their limits of resolution, a multiple length
scale characterization of the pore space is proposed, that is based on a combination of the results from all of
these methods. The approach is demonstrated using samples of Bentheimer and Röttbacher sandstone.
Additionally, we compare the potential of SIP to provide a pore size distribution with other commonly used
methods (MIP, NMR). The limits of resolution of SIP depend on the usable frequency range (between
0.002 Hz and 100 Hz). The methods with similar resolution show a similar behavior of the cumulative pore
volume distribution in the overlapping pore size range. We assume that μ-CT and NMR provide the pore
body size while MIP and SIP characterize the pore throat size. Our study shows that a good agreement
between the pore radii distributions can only be achieved if the curves are adjusted considering the
resolution and pore volume in the relevant range of pore radii. The MIP curve with the widest range in
resolution should be used as reference.
Keywords: Pore Space Analysis, Joint Interpretation, Fractal Dimension, Spectral Induced Polarization
**1 Introduction**
Transport and storage properties of reservoir rocks are determined by the size and arrangement of the pores.
In this paper we use the term geometry to refer to the relevant pore sizes, such as the pore throat radius,
pore body radius, body to throat ratio, shape of the pore, and pore volume corresponding to certain pore
radius. Different methods have been developed to determine the pore size distribution of rocks. These
methods are based on different physical principles. Therefore, it can be expected that the methods
recognize different geometries and sizes. Additionally, the ranges of pore sizes that are resolved by the
methods are different (Meyer et al., 1997). Rouquerol et al. (1994) stated in the conclusions of their
recommendations for the characterization of porous solids that no experimental method provides the
absolute value of parameters such as porosity, pore size, surface area, and surface roughness. It should be
noted that these parameters indicate a fractal nature. That means that the value of the parameter depends on
the spatial resolution of the method.
An enhanced pore space analysis using different methods should be able to provide a better description of
the pore space over a wide range of pore sizes. Our study of pore space analysis is based on the following
methods: micro computed tomography (μ-CT), mercury intrusion porosimetry (MIP), nuclear magnetic
resonance (NMR), and spectral induced polarization (SIP). The first three methods can be regarded as
standard methods to derive a pore size distribution. Since these methods can reveal the inner structure of
the rocks, they are widely applied in geosciences (e.g. Halisch et al., 2016b, Mees et al., 2003,
Behroozmand et al., 2015, Weller et al., 2015). The main aim of our paper is to integrate an electrical
method in this study. Electrical conductivity and polarizability (or real and imaginary part of electrical
conductivity) are fundamental physical properties of porous materials. The SIP method measures the low-
frequency electric behavior of rocks and soil material that can be efficiently represented by a complex
electric conductivity (e.g. Slater and Lesmes, 2002). The electric properties of a porous material depend to
a large extent on key parameters including the porosity, the grain and pore size distribution, the specific
internal surface, the tortuosity, the saturation and the chemical composition of the pore-filling fluids. SIP is
a non-destructive method that can be applied to characterize the geometry of the pore system. Müller-
Huber et al. (2018) proposed the integration of SIP in a combined interpretation with NMR and MIP
measurements for carbonate rocks in order to use the partly complementary information of each method.
The SIP method is used to explore correlations between parameters derived from complex conductivity
spectra and specific pore space properties. We go a step ahead and compare directly the pore size
distributions derived from the different methods. Procedures to derive pore size distributions from induced
polarization (IP) data have been proposed only recently (Florsch et al., 2014; Revil et al., 2014; Niu and
Zhang 2017; Zhang et al., 2017).
We are aware that further methods can be applied for the characterization of pore size distribution, e.g.
synchrotron-radiation-based computed tomography (Peth et al., 2008), focused ion beam tomography
(Keller et al., 2011), transmission electron microscopy (Gaboreau et al., 2012), scanning electron
microscopy (SEM), [14]C labeled methylmethacrylate method (Kelokaski et al., 2005), and gas adsorption
and desorption method (BET) (Avnir and Jaroniec, 1989).
Our study presents an approach to describe and quantify the pore space of porous material by combining
the results of methods with different resolution. Samples of Bentheimer and Röttbacher sandstone are
investigated by µ-CT, MIP, NMR, and SIP. Each method provides the pore size distribution in a limited
range of resolution. It is not our intention to combine the data of the different methods in a joint inversion
to get a more reliable pore size distribution as proposed by Niu and Zhang (2017). We prefer to compare
the resulting pore size distributions to each other to get two different pore radii distributions, one for the
pore body radius and one for the pore throat radius. The comparison of the two curves enables the
determination of the ratio between pore body and pore throat radius. A joint inversion that ignores the
difference between pore body and pore throat provides a simplified model that ignores the complexity of
pore space geometry.
Considering the fractal nature of pore space geometry an attempt is made to determine the fractal
dimension of the pore volume distribution for the two investigated samples. The fractal dimension is a
useful parameter for up-and downscaling of geometrical quantities. Zhang and Weller (2014) investigated
the fractal behavior of the pore volume distribution by capillary pressure curves and NMR $T_2$ distributions
of sandstones. Considering the differences in fractal dimension resulting from the two methods, they
concluded a differentiation into surface dimension and volume dimension. Additionally, the fractal
dimension is used in methods of permeability prediction (e.g. Pape et al., 2009).
**2 Theory**
The pore size distribution resulting from different methods has to be compared and evaluated. We prefer a
comparison based on the cumulative volume fraction of pores $V_c$, which is expressed by
$$V_c = \frac{V(<r)}{V_p},$$
(1)

with $V_p$ being the total pore volume, and $V(<r)$ the cumulative volume of pores with radii less than $r$. A
graph displaying the logarithm of $V_c$ versus the logarithm of the pore radius offers the advantage that the
slope of the curves is related to the fractal dimension of the pore volume (Zhang and Weller, 2014).
Fractal theory is applied to describe the structure of geometric objects (Mandelbrot, 1977, 1983). At
molecular size and microscopic range, surfaces of most materials including those of natural rocks show
irregularities and defects that appear to be self-similar upon variation of resolution (Avnir et. al, 1984). A
self-similar object is characterized by similar structures at different scales. The regularity of self-similar
structures can be quantified by the parameter of fractal dimension $D$. Pape et al. (1982) first proposed a
fractal model (the so called 'pigeon-hole model' or 'Clausthal Concept') for the geometry of rock pores.
Fractal dimension describes the size of geometric objects as a function of resolution. This parameter has
proved to be useful in the comparison of different methods that determine distributions of pores in
sandstones and carbonates (e.g. Zhang and Weller, 2014, Ding et al., 2017).
From MIP, the entry sizes of pores and cavities, which is referred to as pore throat radius $r_t$, can be
determined according to the Washburn-equation (Washburn, 1921)
$$r_t = -\frac{2 \cdot \gamma \cdot \cos\theta}{P_c},$$
(2)

with $\gamma = 0.48$ N/m being the surface tension of mercury, $\theta = 140°$ the contact angle between mercury and
the solid minerals, and $P_c$ the pressure of the liquid mercury that is referred to as capillary pressure.
Starting with low pressure, the pores with larger pore throats are filled with mercury. While increasing the
pressure, the pores with smaller throats are filled. Reaching a certain pressure level $P_c$, a cumulative
volume of mercury ($V_{Hg}$) has intruded into the sample that corresponds to the pore volume being accessible
by pore throat radii larger or equal $r_t$ according to Eq. (2). Figure 1 shows a 2D image of the pore space of
sample BH5-2 (information is given in Section 3) indicating the pore throat radius $r_t$ as measured by MIP
by red arrows. Fluid flow properties, and hence the injection pressure of mercury, solely depends upon the
narrowest pore diameter in the flow path that corresponds to the pore throat diameter. The cumulative
volume of mercury $V_{Hg}$ corresponds to the pore volume $V(>r_t)$. It should be noted that the volume of larger
pores, which are shielded by narrower throats, is attributed to the pore throat radius (e.g. Kruschwitz et al.,
2016). Knowing the total pore volume $V_p$, the saturation of the sample with mercury $S_{Hg}$ can be determined.
A conventional capillary pressure curve displays the relationship between the saturation of the sample with
mercury $S_{Hg}$ as a function of capillary pressure $P_c$ (e.g. Thomeer, 1960). Using the following simple
transformations

$$S_{Hg} = \frac{V_{Hg}}{V_p} = \frac{V(>r_t)}{V_p} = \frac{V_p - V(<r_t)}{V_p} = 1 - V_c , \qquad (3)$$

the cumulative volume fraction of pores $V_c$ as defined in Eq. (1) can be determined as a function of $r_t$.
The NMR relaxometry experiment records the decay of transversal magnetization. The measured
transversal decay curve is decomposed in a distribution of relaxation times $b(T_2)$. The individual relaxation
time $T_2$ is attributed to a pore space with a certain surface to volume ratio $A/V$ by

$$\frac{1}{T_2} = \rho\left(\frac{A}{V}\right), \qquad (4)$$

with $\rho$ being the surface relaxivity. Considering that for a capillary tube model with cylindrical pores of
radius $r$, the surface to volume ratio equals $2/r$, we get the following linear relationship between pore radius
$r$ and relaxation time $T_2$ (e.g. Kleinberg, 1996):

$$r = 2\rho T_2 . \qquad (5)$$

It should be noted that the NMR method resolves the radius $r_b$ that corresponds to the maximal distance to
the pore wall. It can be represented by the pore radius of the largest sphere that can be placed inside this
pore as shown in Figure 1.
Another approach to derive a pore size distribution is based on the SIP method. Relations between grain or
pore size and IP parameters have been reported in a variety of studies (e.g. Slater and Lesmes, 2002; Scott
and Barker, 2003; Binley et al., 2005; Leroy et al., 2008; Revil and Florsch, 2010). Polarization effects of
natural material are caused by different charging and discharging processes of some polarizing elements
such as grain surface, pore throat, membrane, and electrical double layer. Following an approach proposed
by Schwarz (1992), the complex conductivity of an individual polarization element can be presented by a
Debye model. It is assumed that the recorded spectra result from a superposition of polarization processes
characterized by different relaxation times. This approach has been adopted to generate synthetic spectra of
electrical conductivity from distributions of grain sizes (e.g. Revil and Florsch, 2010) or pore sizes (e.g.
Niu and Zhang, 2017).
A decomposition of the spectra is needed to derive the relaxation time distribution. Florsch et al. (2014)
demonstrated that a variety of models can be used as kernel for the decomposition of the spectra. Revil et al.
(2014) compare the results of Debye and Warburg decomposition. Their argumentation, which is based on
mechanistic grain size models describing the polarization of charged colloidal particles and granular
material, supports the application of the Warburg decomposition that results in a narrower distribution of
polarization length scales. It should be noted that a uniform grain size does not automatically generate a
uniform pore size. Besides it can be clearly seen by the scanning electrode microscopy images that the
investigated sandstones feature a distinct range of both, grain and pore (throat) sizes. Considering that the
pore size and not the grain size controls the polarization of sandstones, as observed by different authors (e.g.
Scott and Barker, 2003; Niu and Revil, 2016), a wider distribution of length scales can be expected.
According to our opinion, there are no clear indications for superiority of the Warburg decomposition. Up
to now, a theoretical model that confirms the validity of the Warburg model in describing the polarization
of a simple pore space geometry has not been presented. Therefore, we prefer to use the Debye
decomposition, which has proved to be a useful tool in the processing of IP data in both time and frequency
domain (e.g. Terasov and Titov, 2007; Weigand and Kemna, 2016). The algorithm described by Nordsiek
and Weller (2008) provides the electrical relaxation time distribution as well as the total chargeability from
complex conductivity spectra.
According to the assumption that the electrical relaxation time and pore size are related to each other, the
specific chargeability at a certain relaxation time corresponds to the pore volume attributed to a certain pore
size, and the total chargeability is attributed to the total pore volume of the sample. The volume fraction $V_c$
corresponds to the ratio of cumulative chargeability to total chargeability. To transform the relaxation time
distribution into a pore size distribution, we adopt the approach proposed by Schwarz (1962) and applied
by Revil et al. (2012) for the Stern layer polarization model:
$$r = \sqrt{2\tau D_{(+)}} \, , \tag{6}$$

with $D_{(+)}$ being the diffusion coefficient of the counter-ions in the Stern layer and $\tau$ being the relaxation
time. Originally, this equation describes the relation between the radius of spherical particles in an
electrolyte solution and the resulting relaxation time. Though it remains arguable whether the radius of
spherical grains can be simply replaced by the pore radius (Weller et al., 2016), we generally follow this
approach. Additionally, we assume a constant diffusion coefficient $D_{(+)} = 3.8 \times 10^{-12}$ m²/s as proposed by
Revil (2013).
The signal amplitude at a given relaxation time corresponds to the pore volume related to the pore radius
determined by Eq. (6). Considering the experience that the polarization is related to the specific surface
area per unit pore volume (e.g. Weller et al., 2010), we assume that the IP signals are caused by the ion-
selected active zones in the narrow pores that are comparable with the pore throats. Their size is quantified
by the pore throat radius $r_t$. Following the procedure proposed by Zhang et al. (2017), the cumulative
volume fraction $V_c$ corresponds to the ratio of cumulative chargeability to total chargeability. Considering
the restricted range of pore radii (0.1-25 μm) resolved by SIP, a correction of the maximum $V_c$ becomes
necessary.
**3 Samples and methods**
For this study, two different sandstone samples have been used: first, a Bentheimer sandstone, sample
BH5-2. The shallow-marine Bentheimer sandstone was deposited during the Early Cretaceous (roughly 140
million years ago) and forms an important reservoir rock for petroleum (Dubelaar et al., 2015). This
sandstone is widely used for systematic core analysis due its simple mineralogy and the quite homogeneous
and well-connected pore space. It is composed out of 92% quartz, contains some feldspar and about 2.5
vol.-% of kaolinite (Peksa et al., 2015), which is a direct alteration product of the potassium-bearing
feldspar minerals. Accordingly, surface area as well as surface relaxivity values are mostly controlled by
the kaolinite for this rock.
Secondly, a Röttbacher sandstone, sample RÖ10B, has been used. The Röttbacher sandstone is a fine-
grained, more muscovite-illite containing, and rather homogeneous material that was deposited during the
Lower Triassic era (roughly 250 million years ago). It is suitable for solid stonework and has been widely
used as building material for facades as well as for indoor and outdoor flooring. The Röttbacher sandstone
was included in a study on the relationship of pore throat sizes and SIP relaxation times reported by
Kruschwitz et al. (2016). This sandstone consists mostly of quartz, but features a higher amount of clay
minerals than the Bentheimer sample. Additionally, Fe-bearing minerals (e.g. haematite) have been formed
during its arid depositional environment, giving this sandstone a distinct reddish color. Accordingly,
surface area as well as surface relaxivity are dominated by the clay and Fe-bearing minerals and should be
significantly different than for the BH5-2 sample.
The experimental methods used in this study include digital image analysis (DIA) based upon micro
computed tomography (μ-CT), mercury intrusion porosimetry (MIP), nuclear magnetic resonance (NMR),
and spectral induced polarization (SIP).
For this study, a nanotom S 180 X-ray μ-CT equipment (GE sensing and inspection technologies) has been
used. The sample size for μ-CT scanning is 2mm diameter and 4 mm length. For pore network separation, a
combination of manual thresholding and watershed algorithms has been applied to achieve the qualitatively
best separated pore space. Additionally, separation results have been cross checked with the images of
scanning electrode microscopy (SEM). More details on the DIA workflow can be found in Halisch et al.
(2016). The DIA of the 3-D μ-CT data sets provide for each individual pore the volume and the pore radius
of the largest sphere that can be placed inside this pore (maximum inscribed sphere method, e.g. Silin and
Patzek, 2006) as indicated by the blue circles in Figure 1. Note that Figure 1 displays a 2-D slice with
circles. The DIA is performed in 3-D volumes and provides spheres. The resulting equivalent pore radius is
referred to as pore body radius $r_b$. Though the true extent of the pore is not caught properly, the derived $r_b$
from DIA is a good estimate of the average radius. Adding up the pore volumes starting with the lowest
pore radius yields the cumulative volume fraction of pores $V_c$ (Eq. (1)) as a function of the pore body radius
$r_b$. The µ-CT method can only resolve the part of the pore space with pore sizes larger than the spatial
resolution of the 3D image. Considering a voxel size of 1.75 µm of the 3D data set, and a minimum
extension of pores of two voxels in one direction, which can be separated by the algorithm, a minimum
pore size of 3.5 µm (or minimum pore radius of 1.75 µm) has to be regarded, as for this study, the CT
resolution limit is 1.75 µm. Therefore, the pore volume determined by µ-CT does not take into account the
pore space with radii smaller than 1.75 µm.

The MIP experiments have been conducted with the PASCAL 140/440 instrument from Thermo Fisher
(Mancuso et al., 2012), which covers a pressure range between 0.015 MPa and 400 MPa corresponding to a
pore throat radius range from (at best) 1.8 nm to 55 µm. The samples have been evacuated before the MIP
experiment.

The NMR experiments have been performed with a Magritek Rock Core Analyzer equipment operating at a
Larmor frequency of 2 MHz at room temperature (~ 20°C) and ambient pressure. After drying at 105°C for
more than 24 hours in vacuum, the samples have been fully saturated with tap water with a conductivity of
about 25 mS/m. NMR measurements can be calibrated to get the porosity of the sample. The early time
decay signal corresponds to the total water content. The range of resolved pore body radii depends on the
used value of surface relaxivity. The amplitude $b$ attributed to an individual relaxation time $T_2$ is related to
the volume fraction of pores with the respective pore radius. Considering the larger pores, the resulting
radius corresponds to $r_b$. The smaller pore throats with lower volume yield a lower signal at shorter
relaxation times. The cumulative volume fraction of pores $V_c$ is determined by adding up the individual $b$
values starting from the smallest relaxation time and normalizing to the total sum of all $b$ values.

Complex conductivity spectra were recorded using a four-electrode sample holder as described by Schleifer
et al. (2002). The spectra were acquired with the impedance spectrometer ZEL-SIP04 (Zimmerman et al.,
2008) in a frequency range between 0.002 Hz and 45 kHz at a constant temperature of about 20 °C.
Considering that the complex conductivity spectra are affected by electromagnetic coupling effects,
Maxwell Wagner relaxation and dielectric effects at higher frequencies and by a lower signal to noise ratio
for lower frequencies, we focus on the frequency range between 0.01 Hz and 100 Hz. The samples were
fully saturated with a sodium-chloride solution with a conductivity of 100 mS/m. At least two
measurements were performed for each sample to verify the repeatability. Considering the limited
frequency interval, the SIP method solely resolves a range of pore radii that depends on the diffusion
coefficient. Hence, using $D_{(+)} = 3.8\times10^{-12}$ m²/s in Eq. (6), we get a range of pore radii between 0.1 µm and
10 µm. Smaller pore sizes are hidden by Maxwell Wagner relaxation and dielectric effects that are not
easily related to pore geometry.

Permeability measurements have been performed by using a steady-state gas permeameter (manufactured
by Westphal Mechanik, Celle, Germany), using nitrogen as the flowing fluid. This device features a so
called "Fancher-type" core holder as described by Rieckmann (1970). With this special type of core holder,
significantly lower confining pressures are needed than by using a conventional "Hassler-type" core holder
(12 bar for the "Fancher-type" core holder versus min. 35 - 50 bar for the "Hassler-type" core holder),
leading to much less initial mechanical influence (compaction) upon the sample material. Measurements
have been derived under steady-state flow conditions with accordingly low flow rates in range from 3 to 5
ml/min, leading to measured pressure drops in range from 2 to 7 mbar from sample inlet to outlet. The
derived apparent permeability values have been corrected, to address the Klinkenberg-effect of gas slippage
(Klinkenberg, 1941; API, 1998). Due to the usage of a steady-state technique with low gas flow rates,
correction of the Forchheimer effect of inertial resistance can be neglected (API, 1998).
**4 Results**
**4.1 Petrophysical properties**
Figure 2 (A and C) gives 2-D impressions of the pore system of the Bentheimer sandstone sample. The
pore space in general is very well connected, featuring many large and open pores (Fig.2, A & C, blue
arrows) and can be described as a classical pore body – pore throat – pore body system. Small pores are
mostly found within the clayey agglomerations, which act as (macro) pore filling material (Fig. 2, A & C,
red arrows) and which are homogeneously distributed throughout the sample material. Figure 2 E gives an
impression of the 3-D pore distribution of this sandstone, derived by µ-CT image processing. This
favorable structure is directly reflected by the petrophysical properties of this sandstone. The sample
investigated in our study is characterized by a porosity of 0.238 measured by MIP, a gas-permeability of
$4.25\times10^{-13}$ m² determined by steady-state permeameter (manufactured by Westphal Präzisionstechnik) with
a Fancher type core holder using nitrogen as the flowing fluid, and a specific surface area of 0.3 m²/g
determined by nitrogen adsorption method.
Figure 2 (B and D) shows the pore space of the Röttbacher sandstone sample from 2-D imaging techniques.
Though the (large) pore space is similar structured as it is for the Bentheimer (pore body-throat-body
system, Fig. 2, B & D, blue arrows), it is generally reduced (cemented) by clay minerals and features a
significantly higher amount of small pores within (Fig. 2, B & D, red arrows). Accordingly, pore space
related petrophysical properties classify a more compact rock, which is supported by the 3-D pore
distribution, derived by µ-CT image processing (Fig. 2, F). The sample used for this study features a
porosity of 0.166 measured by MIP, which is lower than for the Bentheimer sandstone. The gas-
permeability is $3.45 \times 10^{-14}$ m², which is less than 10 % of the value determined for the Bentheimer
sandstone. The specific surface area has been measured with 1.98 m²/g and is hence nearly seven times
larger than for sample BH5-2, clearly underlining the impact of the clay content. The petrophysical
parameters for both samples are compiled in Table 1, whereas results from X-ray fluorescence analysis are
summarized in Table 2, regarding the most important chemical components of both sandstones that have
been used for this study.
**4.2 Pore volume fraction**
We applied the methods µ-CT, MIP, NMR, and SIP to get insight into the pore radius distribution of the
Bentheimer sandstone sample BH5-2. Figure 3 displays the resolved porosity $\phi_r$ as a function of pore radius
for µ-CT and MIP data. The cumulative pore volume while progressing from larger to smaller pores $V(>r)$
is normalized to the total volume of the sample $V_s$ and results in the resolved porosity
$$\phi_r = \frac{V(>r)}{V_s},$$  (7)
which reaches the true porosity $\phi$ as threshold value for $r$ approaching zero.
As shown in Figure 3, the µ-CT method identifies the largest pores with pore body radii of about 100 µm.
The resolved porosity $\phi_r$ reaches a value of 0.184 at the limit of resolution of the µ-CT method ($r_b$=1.75
µm). The nearly horizontal curve progression for $r < 17$ µm indicates that effectively no significant volume
of pores with radii lower than 17 µm was detected or quantified by µ-CT and DIA, respectively.
Accordingly, only µ-CT data for r > 17 µm will be taken into account for further analysis.
The MIP identifies the largest pore throats with a radius of about 30 µm. Reaching the limit of resolution of
the MIP, the resolved porosity approaches asymptotically the threshold value of 0.238. Though both
methods µ-CT and MIP yield the pore radius without any adjustable scaling factor, we observe differences
between the two curves $\phi_r(r)$ in Figure 3.
The Röttbacher sample was scanned with resolution 1.5 µm by µ-CT. As shown in Figure 4, the µ-CT
method identifies the largest pores with pore body radii of about 90 µm. The resolved porosity $\phi_r$ reaches a
value of 0.106 at the limit of resolution of the µ-CT method ($r_b$=1.5 µm). As observed for the Bentheimer
sandstone, the nearly horizontal curve progression for $r < 10$ µm indicates that no significant volume of
pores with radii lower than 10 µm were detected or quantified by µ-CT and DIA, respectively. Accordingly,
only µ-CT data for r > 10 µm will be taken into account for further analysis.

### 4.3 Pore radius distribution

The description and quantification of the pore space in three dimensions requires morphological parameters such as length, width, and thickness of individual pore segments. The parameters are extracted by image analysis software from 3-D μ-CT data. We determined the pore length (maximum length of Feret distribution), pore width (minimum width of Feret distribution), and the equivalent diameter of the analyzed pore segment that corresponds to the spherical diameter with equal voxel volume (Schmitt et al., 2016). The minima, maxima, and mean values of the geometrical parameters derived from μ-CT data of the two samples are compiled in Table 3.

The procedures described above result in an individual curve displaying the logarithm of $V_c$ versus the logarithm of the pore radius for each method.

For Bentheimer sandstone, applying the transformation in Eq. (3) for the MIP data and assuming a true porosity of 0.238, the cumulative volume fraction of pores $V_c$ can be displayed as a function of pore throat radius as shown in Figure 5. The MIP curve gets a fixed position in the plot of Figure 5 without the need for any scaling. It covers a wide range of pore throat radii between 0.0018 and 44.7 μm.

The curves resulting from other methods have to be adjusted considering the limits of the range of pore radii. The maximum of the μ-CT curve corresponds to $V_c = 1$ because no larger pore size has been detected by other methods. The maximum resolved porosity of the sample as detected by MIP reaches 0.238. The porosity determined by μ-CT reaches only 0.184 (Figure 3). This value corresponds to a fraction of 0.773 of the porosity determined by MIP. Therefore, the minimum of the μ-CT curve at the pore radius of 17 μm has to be adjusted at $V_c = 1 - 0.773 = 0.227$, because this fraction of pore volume is related to pore body radii smaller than 17 μm. The shift of the μ-CT curve to larger pore body radii in comparison with MIP is observed in this plot, too.

The $T_2$ relaxation time distribution of sample BH5-2 is plotted in Figure 6. It indicates a distinct maximum at a relaxation time of 330 ms and two weaker maxima at lower relaxation times. The $T_2$ relaxation time distribution is transformed into a curve showing the cumulative intensity as a function of $T_2$. The total intensity is attributed to the total pore volume. The volume fraction $V_c$ corresponds to the ratio of cumulative intensity to total intensity. In order to get the curve $V_c$ as a function of pore radius, the relaxation time $T_2$ has to be transformed into a pore radius using the surface relaxivity ρ as scaling factor in Eq. (5). Since both μ-CT and NMR method are sensitive to the pore body radius, we expect a similar $V_c$ - $r$ - curve in the overlapping range of pore body radii. Assuming a coincidence of the two curves at $V_c = 0.5$, the surface relaxivity is adjusted at $\rho = 54$ μm/s.

The complex conductivity spectra of the Bentheimer sample are displayed in Figure 7. Considering the frequency range between 0.01 and 100 Hz and $D_{(+)} = 3.8 \times 10^{-12}$ m²/s, the relaxation time distribution derived from SIP is attributed to a restricted range of pore radii between 0.1 μm and 10 μm. Assuming that the polarization signals originate from the pore throats, a similarity of pores size distributions resulting

from MIP and SIP can be expected. It should be noted that MIP provides the distribution for a wider range
of pore throat radii. Therefore, we adjust the value of $V_c$ at the maximum radius of the SIP to the
corresponding value for the MIP curve.
As shown in Figure 4 for Röttbacher sandstone, the MIP identifies the largest pore throats with a radius of
about 50 µm. Reaching the limit of resolution of MIP, the resolved porosity gets the value of 0.166.
Applying the transformation in Eq. (3) on the MIP data and assuming a true porosity of 0.166, the
cumulative volume fraction of pores $V_c$ is displayed as a function of pore radius as shown in Figure 8.
We suppose that the MIP method detects the whole pore volume, a porosity of 0.106 recognized by µ-CT
corresponds to 63.9% of the total pore volume. Therefore, the minimum of the µ-CT curve at the pore
radius of 10 µm has to be adjusted at $V_c = 1 – 0.639 = 0.361$, because this fraction of pore volume is related
to pore body radii smaller than 10 µm.
The $T_2$ relaxation time distribution of sample RÖ10B is plotted in Figure 6. It indicates a distinct maximum
at a relaxation time of 170 ms. Non-vanishing signals are observed at relaxation times below 0.1 ms. This is
an indication of the existence of very small pores in the Röttbacher sandstone.
The position of the NMR curve in the plot of Figure 8 depends on the surface relaxivity ρ. A coincidence
with the µ-CT curve at $V_c = 0.5$ requires a surface relaxivity of $\rho = 237$ µm/s for adjusting the NMR curve.
The complex conductivity spectra of the Röttbacher sample are displayed in Figure 7. The processing of the
spectra according to the described algorithm results in the $V_c$ - $r$ – curve as shown in Figure 8. The SIP
curve is fixed at the value $V_c = 0.9$ that has been determined by MIP for the maximum pore radius resolved
by SIP ($r_t = 10$ µm).
**5 Discussion**
Previous studies have compared the $V_c$ - $r$ – curves resulting from different methods (e.g. Zhang and Weller,
2014; Zhang et al., 2017; Ding et al., 2017). The slope of the curves was used to get a fractal dimension. It
became obvious that the distribution curves indicate remarkable differences that are caused by the physical
principles of the used methods. The methods differ with regard to their limits of resolution. The effective
resolution of µ-CT is limited by the voxel size. Larger pores can be easily detected. Nevertheless, even
though the derived image (voxel) resolution is quite high (1.75 µm), both sandstone data sets feature no
significant volume of pore body radii smaller than 10 µm (BH-5) and 17 µm (RÖ-10), respectively. We
assume that this is caused by a complex and sensitive mixture of issues about image resolution, image
quality (phase contrast), reliability of the watershed-algorithm concerning the separation of individual pores,
and hence of the complexity of the pore structure of small pores. The MIP yields the widest range of pore
throat radii. The pore radius is directly related to the pressure. A similarly wide range of pore body radii
can be resolved by NMR. However, the transformation of the NMR transversal relaxation time into a pore
radius requires the surface relaxivity as scaling factor. In a similar way, the transformation of the electrical
relaxation time resulting from SIP into a pore radius is based on a scaling factor that depends on the
diffusion coefficient. Only a restricted range of pore radii can be resolved by SIP.
Beside the range of pore radii, the geometrical extent of the pore radius differs among the methods. μ-CT
enables a geometrical description of the individual pore space considering the shape of the pore. The pore
radius can be determined in different ways. We use the average pore radius as an equivalent for the pore
body radius $r_b$. MIP is sensitive to the pore throat radius $r_t$ that enables the access to larger pores behind the
throat. The NMR relaxation time is related the pore body radius $r_b$. We assume that the IP signals are
caused by the ion-selected active zones in the narrow pores that are comparable with the pore throats.
Regarding the differences of the methods, we present an approach that combines the curves to get more
information on the pore space. Considering the two kinds of pore radii $r_b$ and $r_t$, we use first μ-CT and
NMR to generate a combined curve displaying $V_c$ as a function of $r_b$. In the next step, we link the curves
resulting from MIP and SIP to get a curve showing $V_c$ as a function of the pore throat radius $r_t$.
It is fundamental that the total pore volume (or total porosity) has to be known. The cumulative pore
volume fraction should only consider the pore volume that is resolved in the regarded range of pore radii.
Considering the resolution of μ-CT, only the pore space with radii larger than the voxel size is determined.
The cumulative pore volume fraction at the limit of resolution has to be adjusted to the non-resolved pore
volume. In this way, the μ-CT curve gets a fixed position in the $V_c – r$ plot. Regarding NMR, the relaxation
time $T_2$ has to be transformed into a pore radius according to Eq. (5). The application of Eq. (5) requires the
knowledge of the surface relaxivity ρ, which is the necessary scaling factor that causes a shift of the $V_c$ - $r$
curve along the axis of pore radius. Since NMR method is sensitive to the pore body radius, we expect a
similar $V_c$ - $r$ curve for NMR and μ-CT in the overlapping range of pore body radii. The NMR curve is
shifted along the axis of pore body radii until a good agreement between the two curves is reached. This
procedure enables the determination of the surface relaxivity. The proposed alternative method for the
determination of surface relaxivity considers the reduction of NMR relaxation time $T_2$ caused by high clay
content and iron-bearing minerals (e.g. Keating and Knight, 2010).
MIP is used to generate the curve displaying $V_c$ as a function of $r_t$ over a wide range of pore throat radii.
The SIP curve is fixed at the MIP curve considering the coincidence at the largest pore radius resolved by
SIP.
The two curves representing $V_c$ as a function of both $r_b$ and $r_t$ are displayed in a double logarithmic plot.
The horizontal shift of the two graphs represents the ratio $r_b/r_t$. Additionally, the slope of the curves is
related to the fractal dimension.
The proposed approach in this study results in two pore size distribution curves for the two samples, which
are in good accordance to the general pore space structures as described in section 3 and as visualized in
Figure 2 (A to F). The first curve combines the distributions resulting from μ-CT and NMR. The μ-CT data
provide a pore radius, which is regarded as pore body radius, without any scaling. The scaling of the NMR-
curve provides an estimate of the surface relaxivity. The surface relaxivity of the Bentheimer sample
reaches 54 µm/s, the corresponding value of the Röttbacher sample is with 237 µm/s much higher. The
higher surface relaxivity in comparison with the Bentheimer sample is clearly justified considering the
larger specific surface area (Table 1) and the significantly higher content of clay and iron-bearing minerals
as indicated in Table 2.
The two cumulative pore volume distribution curves for the Röttbacher sample (Figure 8) indicate over the
wide range of pore radii a parallel progression with consistently higher values for the pore body radius (µ-
CT and NMR) in comparison with the pore throat radius (MIP). The horizontal distance of the two curves
yields the ratio $r_b/r_t$. It should be noted that the ratio $r_b/r_t$ may vary with pore sizes. Most studies consider
only a fixed ratio $r_b/r_t$ determined from the dominant pore body size from NMR $r_b$ and the dominant pore
throat size from MIP $r_t$ (e.g. Müller-Huber et al., 2018). Regarding the median pore radii at $V_c = 0.5$, a ratio
$r_b/r_t = 9.13$ is determined. Considering smaller pores, a ratio $r_b/r_t = 12.15$ is indicated at $V_c = 0.05$.
The parallelism of the pore volume distribution curve is less developed for the Bentheimer sample (Figure
4). We observe a clear distance of the two curves in the range of larger pore radii. Regarding the median
pore radii at $V_c = 0.5$, a ratio $r_b/r_t = 2.57$ is determined. For $V_c < 0.2$, the slope of the curves decreases and
smaller distances between the curves are observed. The NMR curve in Figure 5 indicates for $V_c > 0.08$
larger pore radii in comparison with the MIP curve and confirms the relationship $r_b > r_t$. The reverse
behavior in the interval 0.1 µm $< r <$ 0.6 µm is possibly caused by the low volume fraction (3%) attributed
to this range of pore radii. It can be expected that the small amount of water in the small pores causes only
weak signals in the NMR relaxometry.
Beside the distances between the curves the individual slopes are regarded. The slope (*s*) of the curve log
($V_c$) versus log (*r*) is related to the fractal dimension *D* of the pore volume ($D = 3 - s$) (Zhang and Weller,
2014). We observe a varying slope in the investigated range of pore radii for the Bentheimer sample. The
only range of more or less constant slope, which extends from pore radius 0.1 µm to 10 µm, corresponds to
a fractal dimension $D_{MIP} = 2.678$ for MIP, $D_{NMR} = 2.776$ for NMR, and $D_{SIP} = 2.618$ for SIP.
The whole curves of the four methods are non-linear and indicate non-fractal behavior. A Maximum
Likelihood Estimator approach (MLE) might be relevant to extract the underlying scaling parameters
(Rizzo et al., 2017). For example, in the case of the NMR curve of Bentheimer sandstone, the fitting of all
data using the MLE reveals that the log-normal distribution is the most likely distribution with the
estimated parameters $\mu = 3.43$ µm and $\sigma = 0.82$ µm. These two scaling parameters are the logarithmic mean
and logarithmic standard deviation of the pore radius, respectively. We recognize that the resulting mean
radius reaches half value of the effective hydraulic radius ($r_{eff} = 6.97$ µm).
We observe a constant slope of the NMR curve for the Röttbacher sample (Figure 8) in the interval
0.01 µm $< r_b <$ 100 µm. A similar slope is observed for the MIP curve in the interval 0.01 µm $< r_t <$ 10 µm.
Considering the overlapping pore throat radii range between 0.1 µm and 10 µm, a fractal dimension *D* with
values of 2.640 for MIP, and 2.661 for NMR has been determined. The slightly higher slope of the SIP
curve results in a lower value of fractal dimension of $D = 2.533$.
Our approach enables the integration of SIP in the determination of a pore throat size distribution.
Considering the limited frequency range, only a limited range of pore throat radii can be reflected. Using a
fixed diffusion coefficient $D_{(+)} = 3.8 \times 10^{-12}$ m²/s, a range of pore throat radii between 0.1 µm and 10 µm is
resolved. The SIP curve is linked to the MIP curve at $r = 10$ µm. An extension to lower pore radii would
require the integration of higher frequencies. The removal of electromagnetic coupling effects can be one
first step to improve the reliability of complex conductivity spectra for frequencies larger than 100 Hz, but
it should be regarded that smaller pore sizes are hidden by Maxwell Wagner and dielectric relaxation. The
proposed procedure results in a fair agreement between SIP and MIP curves in the overlapping range of
pore throat radius for both the Bentheimer and the Röttbacher sample. In comparison with MIP, a slight
overestimation of $V_c$ is observed for larger pore throat radii and a underestimation for lower pore throat
radii. Considering the two samples of the presented study, the assumption of a constant diffusion
coefficient seems to be justified. Though alternative kernels have not been tested, our study confirms that
the Debye decomposition provides a relaxation time distribution of complex conductivity spectra that can
be transformed in a pore throat size distribution comparable with the resulting curves from MIP. Regarding
the discussion on the most relevant parameter that controls the relaxation time, our assumption that the pore
throat radius is related to the relaxation time is supported by the results.
The investigations by µ-CT, MIP, NMR, and SIP on the sandstone samples have been done in the
laboratory. µ-CT and MIP are methods that can only be applied on rock samples. The potential of these
methods to derive pore size distributions is well acknowledged. NMR and SIP are methods that can also be
performed in boreholes or as field survey. The NMR method has been successfully applied in permeability
prediction at the field scale. A variety of permeability prediction models based on SIP parameters has been
proposed based on laboratory investigations (e.g. Robinson et al., 2018). First tests have demonstrated their
applicability in the field. Most permeability models consider pore size and porosity as the most important
parameters. The evaluation of pore sizes of sediments at the field scale is a challenging task for geophysical
methods. Our laboratory study has demonstrated the potential of SIP in identifying a pore size distribution.
Further investigations with larger sets of samples have to be done to improve the proposed procedure
before the pore size distribution can be extracted from high quality complex conductivity field spectra.
**6 Conclusions**
Pore radii distributions (considering both pore body and pore throat radii) have been determined by
different methods (µ-CT, MIP, NMR, and SIP) for two sandstone samples. The curves presenting the
cumulative distribution of pore volume $V_c$ as a function of pore size have proved to be a suitable tool for
comparison. It becomes obvious that the distribution curves indicate remarkable differences that are based
on the physical principles of the used methods. The methods differ with regard to their limits of resolution.
The effective resolution of μ-CT is limited by the voxel size (1.75 μm). Larger pores can be easily detected,
whereas quantification of small pores and volumes of pores with small radii is severely affected by the
image quality and the image processing algorithms. The MIP yields the widest range of pore radii. The pore
throat radii are directly related to the pressure interval. A similar wide range of pore radii can be achieved
by NMR. However, the transformation of the NMR transversal relaxation time into a pore body radius
requires the surface relaxivity as scaling factor. In a similar way, the transformation of the electrical
relaxation time resulting from SIP into a pore radius is based on a scaling factor that depends on the
diffusion coefficient. Only a restricted range of pore radii (0.1 μm to 10 μm) can be resolved by SIP.
Beside the range of pore radii, the geometrical extent of the pore radius differs among the methods. μ-CT
enables a geometrical description of the individual pore space considering the shape of the pore. The pore
radius can be determined in different ways. We use the average pore radius as an equivalent for the pore
body radius $r_b$. MIP is sensitive to the pore throat radius $r_t$ that enables the access to larger pores behind the
throat. The NMR relaxation time is related to an average pore body radius $r_b$. We assume that the IP signals
are caused by the ion-selected active zones in the narrow pores that are comparable with the pore throats.
Considering the two kinds of pore radii $r_b$ and $r_t$, we use μ-CT and NMR to generate a combined curve
displaying $V_c$ as a function of $r_b$. A good agreement between the two curves is achieved if they coincide at
$V_c = 0.5$. This condition is used to determine the surface relaxivity, which is in good accordance to the
investigated surface area and mineralogy of the sample materials. MIP is used to generate the curve
displaying $V_c$ as a function of $r_t$ over a wide range of pore throat radii. The SIP curve is fixed at the MIP
curve considering the coincidence at the largest pore radius resulting from SIP.
The two curves representing $V_c$ as a function of both $r_b$ and $r_t$ are displayed in a double logarithmic plot.
The horizontal shift of the two graphs represents the ratio $r_b/r_t$. Additionally, the slope of the curves is
related to the fractal dimension.
The investigations on the samples demonstrate that the porosity increases using a method with a higher
resolution. Both porosity and pore volume are parameters that depend on the resolution. The fractal
dimension describes the size of geometric objects as a function of resolution. Therefore, the knowledge of
fractal behavior enables upscaling and downscaling of geometric quantities. The Bentheimer sandstone
sample is characterized by a ratio $r_b/r_t = 2.57$ for the larger pores. A fractal behavior is observed in the
range of pore radii between 0.1 μm and 10 μm with an average $D = 2.69$ determined for the pore volume by
MIP, NMR, and SIP. The Röttbacher sandstone sample indicates with $r_b/r_t = 9.13$ a larger ratio between
pore body radius and pore throat in comparison with the Bentheimer sample. An average fractal dimension
of $D = 2.61$ is determined for the Röttbacher sample.

## Acknowledgements

The authors thank Sven Nordsiek (University Bayreuth) for the Debye decomposition of the SIP data, Dietmar Meinel (BAM, Berlin) for supporting the CT analysis, Carsten Prinz (BAM, Berlin) for providing the MIP data, and Mike Müller-Petke as well as Raphael Dlugosch (both Leibniz Institute for Applied Geophysics, Hanover) for the acquisition of the NMR spectra for this study. Dr. Zeyu Zhang thanks Bundesanstalt für Materialforschung und –prüfung (BAM, Berlin) for the Adolf-Martens-Fellowship that enabled his stay in Germany for the experimental research.

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

**Table 1: Petrophysical properties of the samples: porosity $\phi$, permeability $K$, specific surface area $S_m$, formation**
**factor $F$, dominant pore radius $r_{dom}$, effective pore radius $r_{eff}$, the ratio $r_b / r_t$, fractal dimensions determined from**
**mercury intrusion porosimetry $D_{MIP}$, nuclear magnetic resonance $D_{NMR}$, and spectral induced polarization $D_{SIP}$,**
**the surface relaxivity $\rho$, and the Diffusion coefficient $D_{(+)}$.**

|  | unit | BH5-2 | RÖ10B |
|---|---|---|---|
| Porosity (triple weighing) |  | 0.238 | 0.159 |
| Porosity (μ-CT) |  | 0.184 | 0.106 |
| Porosity (MIP) |  | 0.238 | 0.166 |
| Permeability $K$ | m² | $4.25\times10^{-13}$ | $3.45\times10^{-14}$ |
| Specific surface area | m²/g | 0.30 | 1.98 |
| Formation factor $F$ |  | 14.3 | 11.3 |
| $r_{dom}$ (MIP) | μm | 11.4 | 4.9 |
| $r_{eff} = (8FK)^{0.5}$ | μm | 6.97 | 1.77 |
| $r_b / r_t$ |  | 2.57 | 9.13 |
| $D_{MIP}$ |  | 2.678 | 2.640 |
| $D_{NMR}$ |  | 2.776 | 2.661 |
| $D_{SIP}$ |  | 2.618 | 2.533 |
| Surface relaxivity $\rho$ | μm/s | 54 | 237 |
| Diffusion coefficient $D_{(+)}$ | m²/s | $3.8\times10^{-12}$ | $3.8\times10^{-12}$ |

**Table 2: Chemical components of the samples from X-ray Fluorescence analysis.**

| | Selected chemical components from X-Ray Fluorescence [weight-%] | | | | | | |
|---|---|---|---|---|---|---|---|
| Sample | $SiO_2$ | $TiO_2$ | $Al_2O_3$ | $Fe_2O_3$ | CaO | $Na_2O$ | $K_2O$ |
| BH5-2 | 97.84 | 0.048 | 1.2 | 0.05 | 0.019 | 0.02 | 0.355 |
| RÖ10B | 87.06 | 0.356 | 6.06 | 1.07 | 0.225 | 0.13 | 3 679 |


**Table 3: Geometrical parameters of individual pores derived from µ-CT data of the two samples.**

| descriptor | Sample | | | | | |
|---|---|---|---|---|---|---|
| | BH5-2 | | | RÖ10B | | |
| | min. [µm] | max. [µm] | mean [µm] | min. [µm] | max. [µm] | mean[µm] |
| equivalent pore diameter | 2.17 | 229.1 | 71.56 | 1.86 | 230.4 | 28.95 |
| Feret length (length 3D) | 1.92 | 537.1 | 161.8 | 1.64 | 416.8 | 56.3 |
| Feret Width (width 3D) | 1.92 | 307.0 | 87.45 | 1.64 | 265.8 | 28.28 |
| Feret breadth (breadth 3D) | 1.75 | 379.4 | 114.7 | 1.5 | 354.8 | 37.99 |
| pore volume [µm³] | 5.36 | 6294270 | 315069 | 3.38 | 6404830 | 64809 |


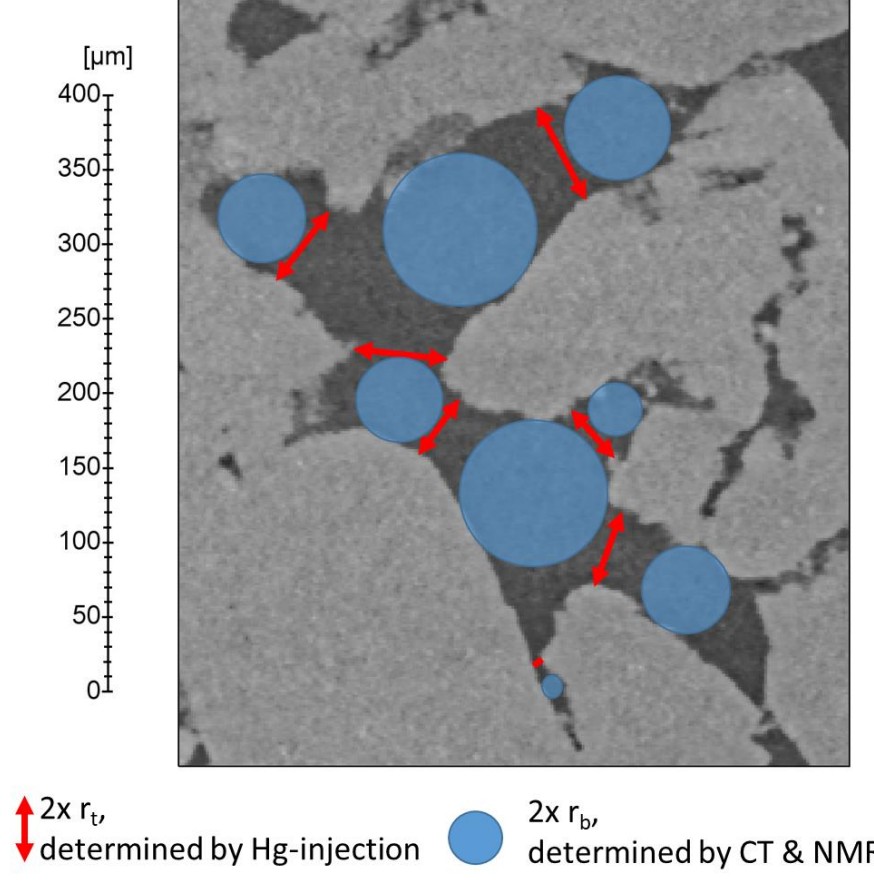

**Figure 1: Zoomed in 2-D slice view of sample BH-5 in order to visualize pore bodies (blue circles, detected by**
**NMR and DIA of μ-CT data) and pore throats (red lines with arrows, detected by MIP).**

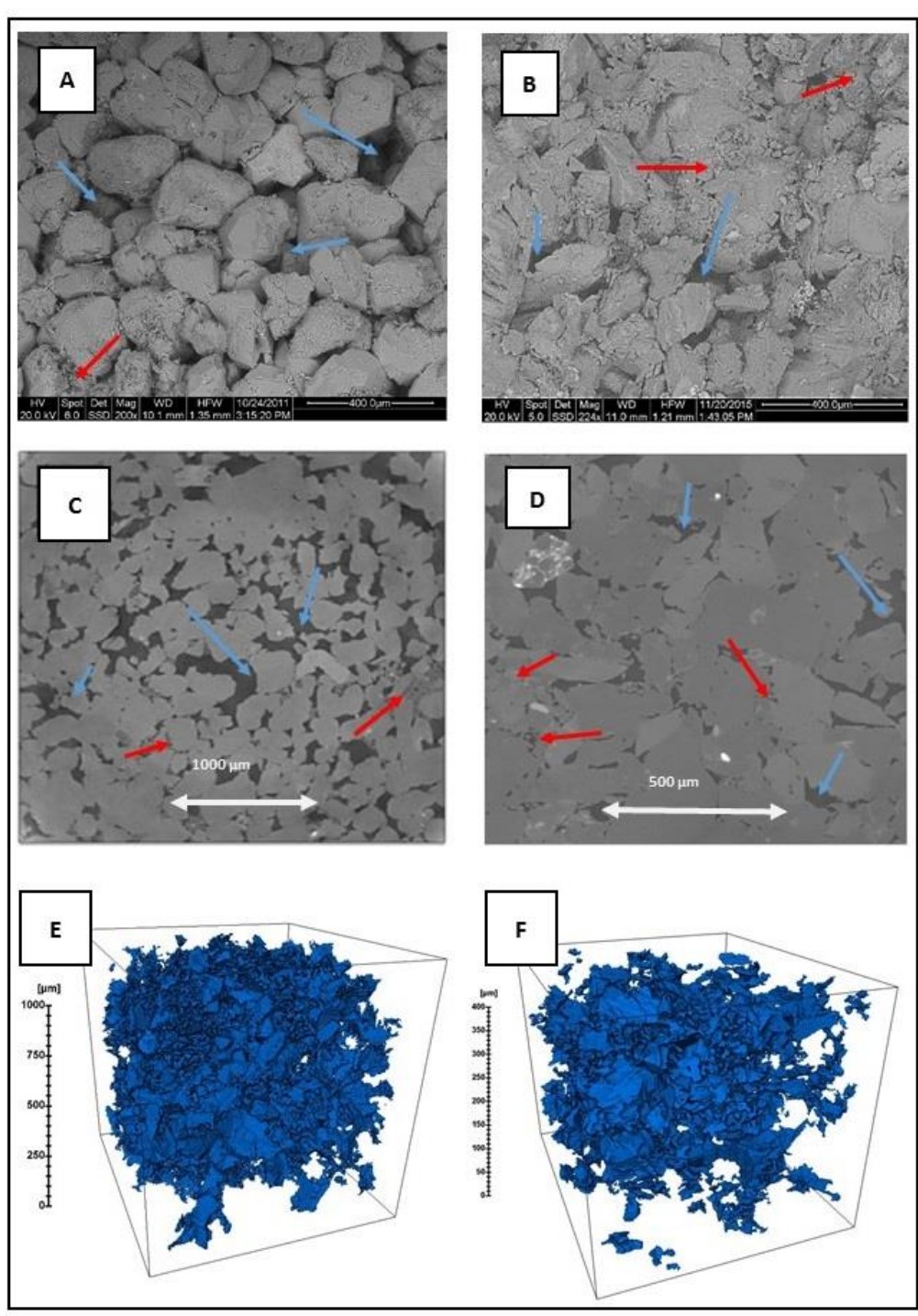

**Figure 2: SEM (A) and 2D (C) and 3D (E) CT views upon the minerals and pore structure of the investigated**
**sample of Bentheimer sandstone, and SEM (B) and 2D (D) and 3D (F) CT views upon the minerals and pore**
**structure of the investigated sample of Röttbacher sandstone. Blue arrows indicate open pore spaces, red arrows**
**indicate clay agglomerations and pore fillings.**

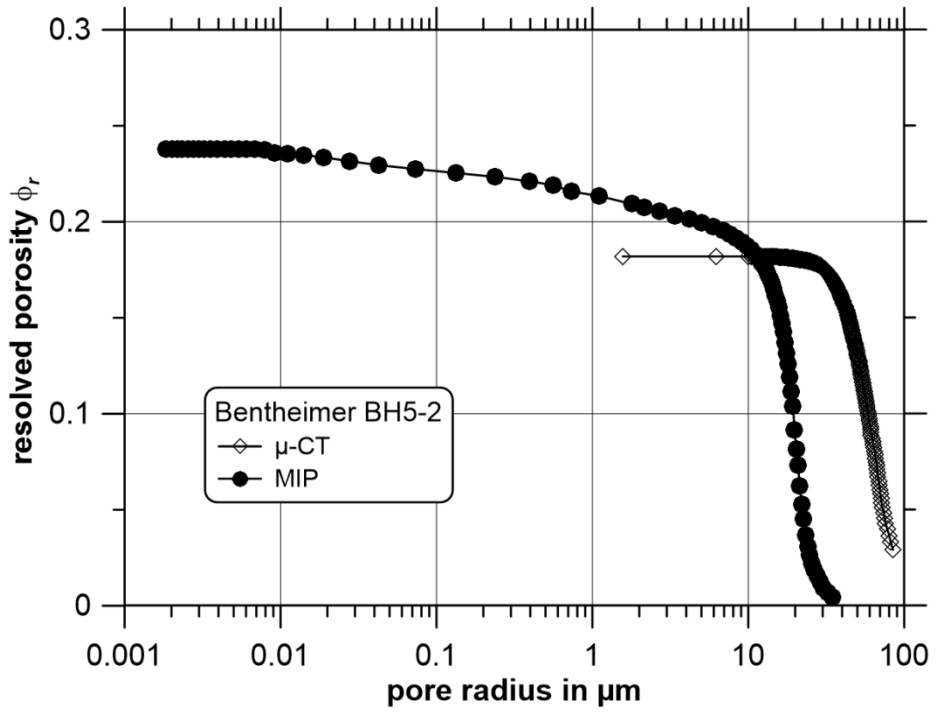

**Figure 3: The recognized porosity and pore size range of Bentheimer sandstone sample BH5-2. The maximum**
**porosity recognized by MIP is 0.238 and the maximum porosity recognized by μ-CT is 0.184.**

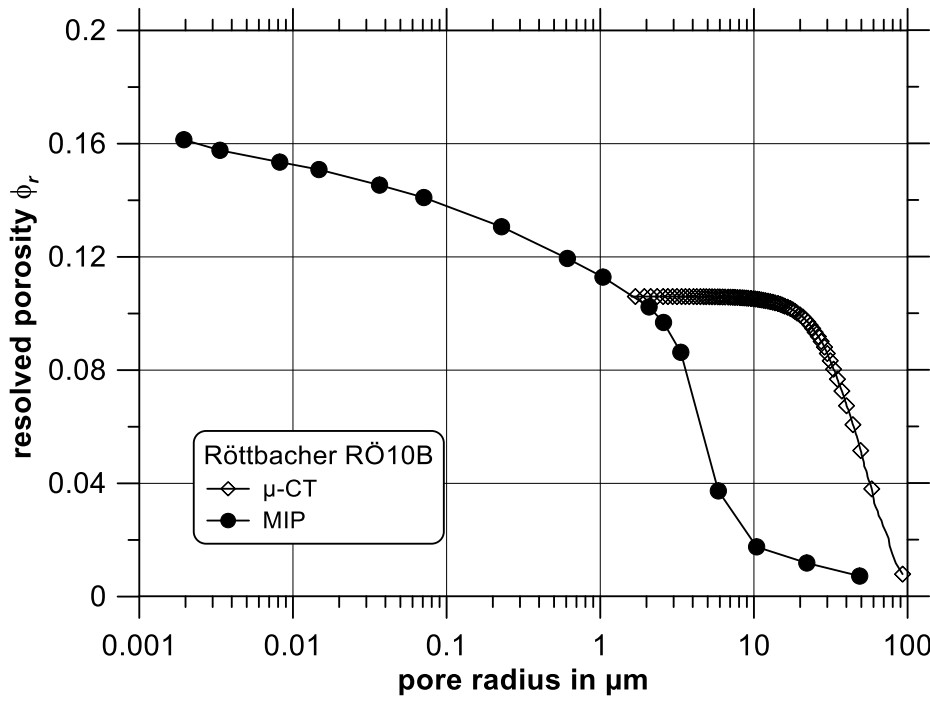

**Figure 4: The recognized porosity and pore size range of Röttbacher sandstone sample RÖ10B. The maximum**
**porosity recognized by MIP is 0.166 and the maximum porosity recognized by μ-CT is 0.106.**

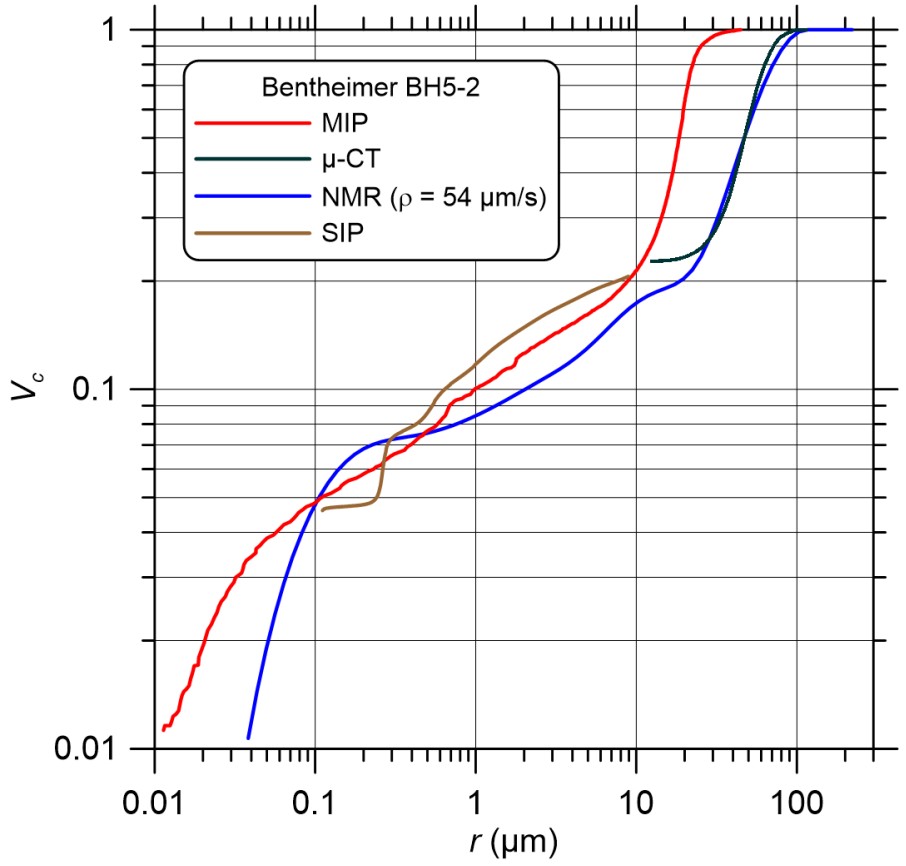

**Figure 5: The comparison of $V_c$-$r$ curves determined from MIP, μ-CT, NMR and SIP for Bentheimer sandstone**
**sample BH5-2.**

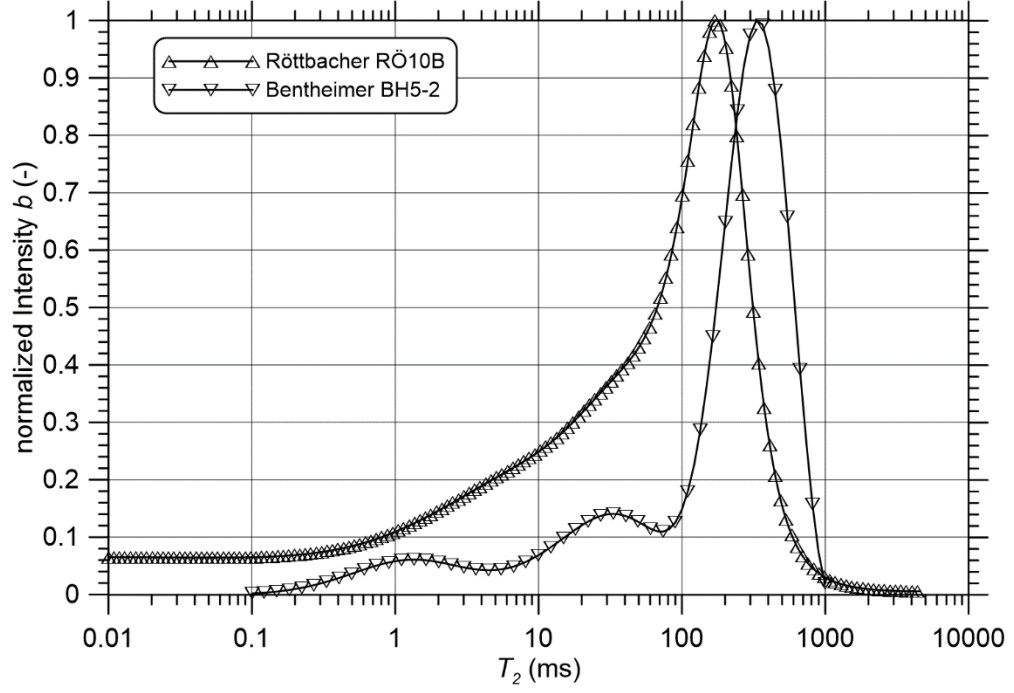

**Figure 6: The NMR $T_2$ relaxation time distributions of samples BH5-2 and RÖ10B.**

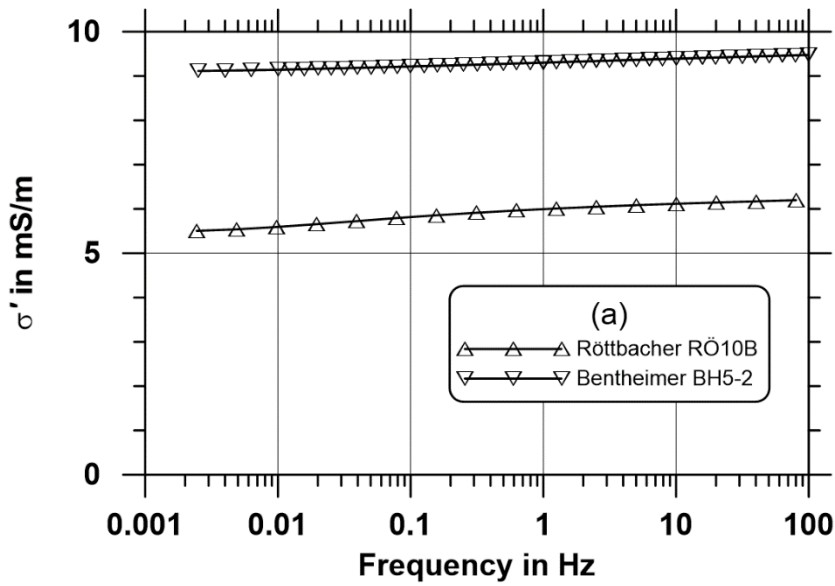


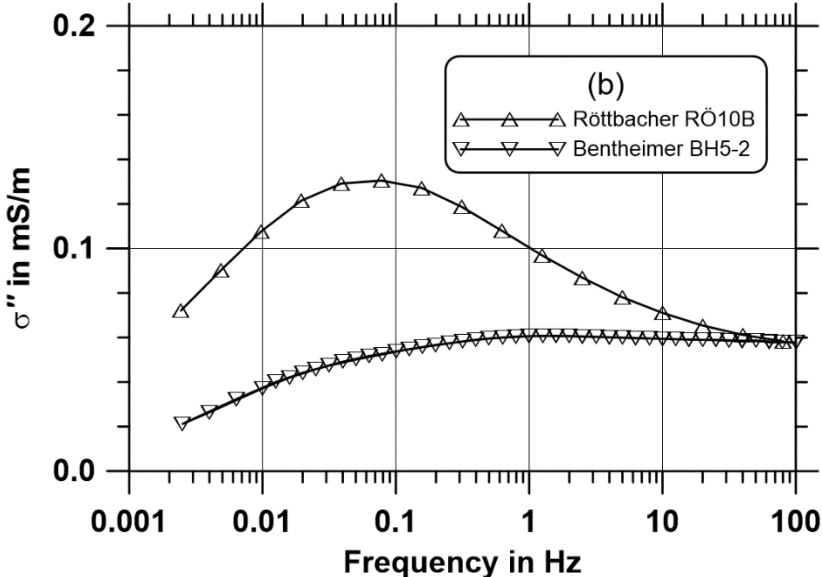

**Figure 7: Measured complex conductivity spectra of samples BH5-2 and RÖ10B. a) real part of conductivity, b)**
**imaginary part of conductivity.**

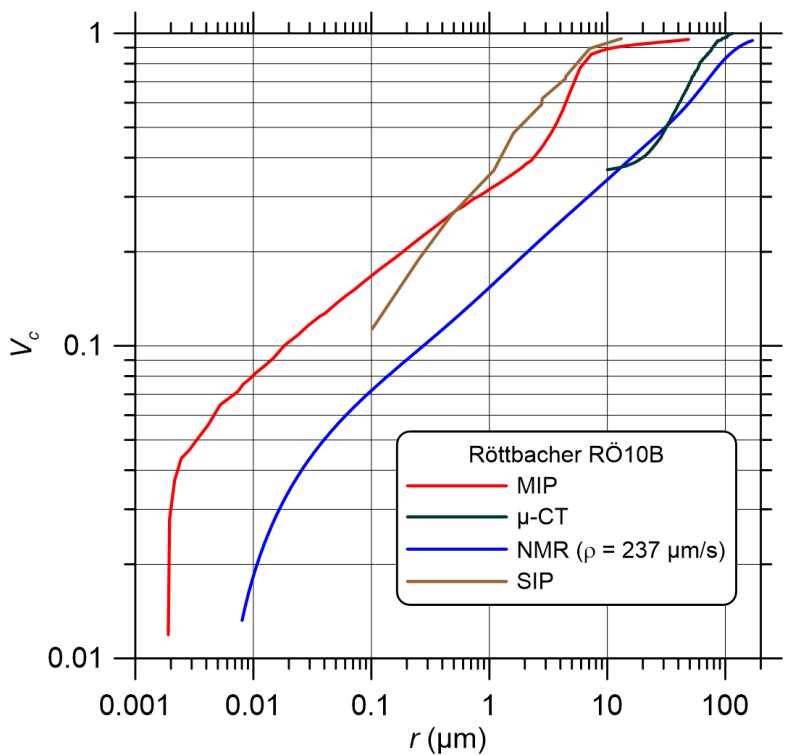

**Figure 8: The comparison of $V_c$-$r$ curves determined from MIP, µ-CT, NMR and SIP for Röttbacher sandstone**
**sample RÖ10B.**