# Peer review of "Enhanced pore space analysis by use of µ-CT, MIP, NMR, and SIP Zeyu Zhang1, Sabine Kruschwitz2,3, Andreas Weller4, Matthias Halisch5 1Southwest Petroleum University, School of Geoscience and Technology, 610500 Chengdu, China 2 Federal Institute for Material Research and Testing (BAM), D-12205 Berlin, Germany 3 Technische Universität Berlin, Institute of Civil Engineering, D-13355 Berlin, Germany 4"

_Solid Earth, 2018_

## Referee Comment (RC1) · Anonymous Referee #1 · 31 May 2018

General comments:

The analysis of pore space geometry in rocks is of great relevance to many areas of earth sciences. In this study, the authors propose to compare the results obtained from different pore space characterization techniques in two sandstones, the Bentheimer and the Rotthbacher.

Though I think this is a great topic to investigate, I find that the work itself does not bring significant value in its present form. A lot of research has already been conducted in this domain and it is difficult to see what new element the paper provides, aside perhaps from the spectral induced polarization part which I unfortunately had a very hard time following. Maybe the work could be augmented with a better review of previous findings and a more thorough extraction of information from the microCT images.

[Figure]

In the abstract, the authors announce that they are going to characterize the pore space geometry, although nothing is done beyond acquiring and tracing various cumulative curves. In the abstract again, the concept of fractals is used but it is unclear (1) whether it is warranted to compute a fractal number from these curves as it is not demonstrated that they represent distributions of objects and (2) what the authors recommend what one should do with that value.

Regarding the work that is done on images and the comparison between different data sets, I wish the authors had provided more information and figures on the image segmentation result as well as the result of the maximum inscribed sphere (MIS) computation. Also, the authors may be aware that such computation (MIS) can be used as a starting point for performing a digital equivalent to MIP which becomes then valuable to compare with the experimental mercury injection curve. In fact, a comparison between experimental MIP and digital MIP on one hand, and between MIS and NMR on the other would have made more sense.

Because the reader does not have access to the state of the images prior to the tracing of the cumulative 'pore size' curve (MIS), it is very difficult to check whether the result is consistent. I have a doubt regarding the offests observed in Figures 2 and 6 and I am wondering if what is plotted for the MIS is really the radius or rather a diameter. Please check. Also I am surprised to see that virtually no 'objects' with dimension smaller than 20 microns was detected in either sandstone considering the image resolution of 1.75 microns per voxel and 1.5 microns per voxel for the Bentheimer and the Rothbacher, respectively.

I don't think that I can speak at length to the SIP part because I am not familiar with it. I would like to see a more intuitive explanation as to why it is appropriate to compare SIP data with a drainage (MIP) curve. Also I don't understand how the data of Figure 5 on frequency-dependent complex conductivity is converted into relaxation times (assuming this is what is being done).

In terms the organization of the paper, I found the figures confusing in the fact that they convey more information than is being discussed at first, forcing the authors to go back and forth when describing their results.

I am convinced the authors have at they disposal a great starting point to a valuable study. The sandstones picked are definitely materials of interest to the community and the high resolution microCT images can certainly be exploited further.

Other comments:

*I think the English could probably be improved (grammar and choice of words mostly)

*In the conclusion, an image resolution of 3.5 microns per voxel is quoted - please decide.

*I did not see the benefit of plotting the curves starting from the smallest injection radii - it puts an emphasis on the fraction where there is less data and also that likely contributes nearly nothing to flow, while dwarfing most of the important information.

*How were the injection steps chosen for the MIP curves? It seems nearly random, and also very sparse in the case of the Rothbacher.

*Table 2 would be easier to look at if some mineral names were added to it.

*The resolution of the microCT images is great, the authors should be able to show much more detail at the grain scale. Have the authors attempted to determine whether the voxel dimension was a true image resolution?
* * *

---

## Referee Comment (RC2) · A. Bubeck (Referee) · 6 Jun 2018

Manuscript No. SE-2018-42

Reviewers' Comments:

The submitted manuscript describes a study that aims to characterize pore size distributions, and the scaling of attributes, for two well-constrained types of sandstone using the following methods: 1) micro CT; 2) mercury intrusion porosimetry; 3) nuclear magnetic resonance; and 4) spectral induced polarization. Using curves of the cumulative volume fraction of pores and pore body radius, the methods are compared.

Generally speaking, the aims of the paper are clear, and data are well-presented. Improving our understanding of the nature of pore space in natural rocks is important to

a broad range of study areas, including petrophysics and rock mechanics, for which a comparison of methodologies will be potentially very useful. It is my view, however, that the manuscript, in its present form, contributes little to our understanding of natural pore variability; resolution cut-offs for these techniques have been discussed previously. Many of the sections need rewriting to avoid overlap and repetition, and the motivation, and context for the study are unclear. For these reasons, it has been a challenge to review as thoroughly as I would like. Some simple restructuring and clarification could improve this but additional work is also necessary. For instance, the authors state early on that they provide a "multiple length scale characterization of pore geometry" - where is this data? A sentence in the conclusion section refers to "$\mu$-CT enables a geometrical description of individual pore space...", but no data are presented. Providing additional geometric data would greatly improve the novelty of the work, and appeal to a broader readership. If the authors lack this data, I recommend that they remove references to geometry throughout. Furthermore, there is virtually no review of existing geological (or other) data derived using the techniques, nor their application, and a scattered discussion of the results towards the end of the manuscript. This needs work before it can be considered for publication.

The authors may decide that additional data, and/or background and discussion is beyond the scope of this work. If this is the case, then considering the broad readership of Solid Earth, they might consider submitting the work to a more subject-specific journal.

As a final note, the manuscript should be given thorough proof-read from a native English speaker; language, grammar, and sentence structure require work before I think it is ready to publish. I have not made specific edits related to these because they are numerous. Below, I have provided section-specific comments for the authors. I hope the authors find them helpful.

MS Specific Comments

0. Abstract The abstract reads fine and, with some minor edits, reflects the main

findings of the manuscript.

Line(s) Comment 39 "...pore space geometry..." Global edit: you do not discuss pore geometry in this work. Please remove reference to it unless you have additional data to add.

1. Introduction This section needs some restructuring, and expansion. For submission to Solid Earth, the work should mention the existing applications of these methods (geological, and others). The authors could consider including a background section to present existing uses of each method, their limitations, and how this data is used. This would improve the framing of the work, and broaden its appeal to different readers. These are some questions I have from this section: a. What are your motives for the study? b. Who are you targeting with this work? Petrophysicists? Is Solid Earth the best place to present this? Porosity, and its effect on fluid flow and the mechanical behaviour of porous rocks is important to a range of study areas, and the subject of extensive study. A (brief) review of some the existing literature would help - e.g. recent uses of x-ray CT to analyse pore volume/geometry/distribution and fluid flow through porous media.

Line(s)

52-57 This isn't much of an introduction for the study. 63-67 I think you could move this to your discussion section and expand on how results from these studies compare to your own. 68-75 This section isn't very clear. You mention 3 separate published NMR studies but no others? It is not immediately clear what the relevance of these studies are to your own motivation, or results. You should add some background for the other methods too. Speaking for micro CT, there are several studies that analyse rock pores that the authors should take a look at, including: Lindquist et al., 2000; Ketcham, 2005; Nakashima and Kamiya, 2007; Takahashi el al., 2016; Schmitt et al., 2016; Saenger et al., 2016; Bubeck et al., 2017; Zhao et al., 2017 (this last one should be particularly helpful in helping you structure and present your work).

76-83 This section is partly repeated. Read through your introduction and keep it concise. 77 Either: add a section that deals with pore geometry specifically, and include additional figures/data, or remove this. 80 ". . .distributions are connected to each other. . ." This is unclear.

2. Methodology

This section is very long and it looks like a mixture of background and methodology, which could be split up. Keep your methodology simple: what did you do, and to what? A number of questions need to be addressed in this section to improve the clarity of your results. a. How many samples did you use in total? b. How many samples did you run for each test? This needs explanation to demonstrate that the results are repeatable. You could add this data to table 1. You should also explain what size samples need to be. c. Does sample size effect the results for any of the methods? It should certainly be considered in micro CT. d. Did you characterize grain size? What role will this play?

Line(s)

91-99 This is background 99 The use of fractals (in geology alone) stretches considerably beyond these references. If you are presenting data on pore scaling relationships, expand this and move to a background section. 100-102 Good. You should move your sample descriptions into this section. 103-108 Tell me what you're doing/using before you tell me the limitations of it. 109-110 How have you performed this analysis, what were the results, and why is it best? More information and figures to explain this process are needed. Why compare with 2D SEM images? 113 ". . . each individual pore. . ." Where are your data for individual pores? You could present data on the proportion of pore body radii (how variable are they within samples? How do your different sandstone types compare?), their geometry and preferred orientation (if present). 114 Do you have references for this technique? 119 You're using MIP specifically for pore throat radius? Be clear about what data each method is providing! 131-132 Try to

explain this more clearly. 139-145 Background. 142 "a capillary model with cylindrical pores, of uniform radius..." - How useful is this? Is it reasonable to assume a linear relationship for natural pores, which may be neither cylindrical, nor uniform? Kleinberg, 1996 applies it to "slit-like pores". You should provide some geometric description of your pore space to support the use of this model. At the VERY least, explain how your results could vary. You could leave this as is here, but add a section to your discussion covering possible limitations of the method and expand on this there. 150-151 "The range of resolved pore radii depends on the used value of surface reflexivity." - You need to explain what value you have chosen, and why? If similar assumptions apply to the use of a single value here, address it in the discussion also. 175 Quantify the "restricted range" 182 Did you have any repeatability issues? Do you think two sufficient? 187-224 This section is a mixture of background, discussion and results, with some repetition. Read through carefully and move elsewhere where appropriate.

3. Sample material This section is a mixture of methodology and results. Does it need to be a separate section? I think it would be helpful to have this information earlier. When you're describing the methods, it will help to know what they are related to. The section could also be much shorter. I recommend you edit the information into your methodology, explaining the number of samples used and their sample sizes

Line(s)

227-228 Cut this. Unnecessary. 244-245 Remove 245-247 Background 251-262 This is your data? It needs to be in the results section.

4. Results The description of the graphs in this section is rather vague and unhelpful - you refer to "differences" between curves but don't describe or quantify these. Equally, stating that something is "larger" or "smaller" is also unhelpful. Quantify your observations!

I suggest changing the section titles here to refer to the parameters measured, i.e. pore volume fraction, pore throat radius, and scaling. Describe the results of these for

each rock sample. This would be easier to follow than describing each rock; it is the technique that should be the focus of this section.

Your fractal data is currently lost in your discussion section. Move it into your results and consider a table that compares the dimensions obtained.

Line(s)

282 "...wide range of pore radii..." State the range measured for each method, with upper and lower limits clearly defined. 289 Explain this more clearly.

5. Discussion

This section is a confusing mixture of results and discussion. You should read through it carefully and remove results to earlier sections.

After reading the discussion, I have a number of questions that can be addressed by restructuring and expanding the discussion. a. What are the implications for the pore size distributions you have obtained? There is no real discussion of the importance of pore size distributions, or their use, in this work... b. What effect do you think your results have for studies of the mechanical behaviour and fluid flow properties of your samples? c. How do your results compare to existing characterisations of these samples - are you offering an improved resolution? How important is it for samples like this? d. How strongly dependent are your resolutions on rock type? How would they vary for other rock types: Limestones/volcanic rocks for example? e. Are the approaches described suitable for a range of rock types/sample sizes/porosities? I would like to see this section discuss the implications of the results more broadly. f. Which of the techniques is best? Which one provides the most, useful, information - and for whom? g. What are the limitations of your findings?

A summary table of the data provided by each method, and resolution would be helpful to readers. Also include a limitations section and use this as a basis to explain which methods should be used for certain applications, and how cautious researchers should

we be when interpreting the results.

6. Conclusions A lot of the material in this section could be moved into the discussion. Add a sentence summarising the importance of the result of this study.

Line(s) 399-407 These are (some of) your conclusions. The rest can be moved.

7. Figures/Tables

Suggested new figures: a. Demonstrate the 3D pore network would be helpful when describing your samples (e.g. Figure 3 in Zhao et al., 2017) b. Segmentation/MIS c. Plot your body radii against cumulative frequency.

Figure 1 No changes suggested. Figure 2 Label the two values (0.238 and 0.184) in the caption. Figure 3 No changes suggested. Figure 4 Label each part as A and B; describe them separately in the caption. Figure 5 No changes suggested. Figure 6 Label values as for figure 2. Figure 7 No changes suggested.

Table 1 Where does your permeability data come from? I recommend you convert these values to m2. Table 2 No changes suggested.

---

## Referee Comment (RC3) · D. Healy (Referee) · 20 Jun 2018

General comments This ms is a comparison of 4 methods for characterising pore space, especially pore size distributions. Data from tests on 2 sandstones are compared. Overall, it is a good idea, quite well presented, and should provide a useful addition to the literature. It's easy to suggest extra tests or analyses but the 4-way comparison stands as is (but see below), and the separation into methods better suited for pore bodies and pore throats is good. I think this paper could be acceptable, subject to moderate revision. My main issues are with the relative lack of quantitative comparison of the methods, and the apparent underlying assumption of power law behaviour. E.g. in Figures 3 and 7, assuming you can make a case for power law behaviour, what is the departure of each line/method from a modelled power law prediction? The paper

does not go anywhere near far enough in this regard, in my opinion.

Specific comments Line 88-90 – much more clarity needed here; a log-log plot of these variables MAY show a straight line, which COULD then be interpreted as power law behaviour. Are you assuming a power law, and therefore fractal behaviour of the data? Perhaps a Maximum Likelihood Estimator approach might be relevant here (Clauset et al., 2009 SIAM Review; Rizzo et al., 2017 Journal of Structural Geology). Line 106-107 – but this limit is rock/ CT scanner/segmentation dependent, right? So add the caveat, 'for this study, the CT resolution limit is . . .'. Line 165 – 'arguable' Line 199 – 'extent' Line 231, 243 – 'mainly'; can you be more specific about the modal proportions of minerals? Line 240, 256 – what method for the porosity estimate? Line 249 – 'depositional' Line 251 – 'shows' Line 384 – 'extent'; 'differs' Line 401 – 'of' not 'on' Line 508 – Table 2; these are not mineral phases, they are chemical components! Line 512/Fig 1 – SEM; what detector, BSE? Say so. Line 520/Fig 2 – I think we need to see 'raw' data for these methods; AND then the 'processed' data using the 'known' porosity. Let the reader judge the data. Line 523/Fig 3 – my point above about assuming power law/fractal behaviour (NB, not the same thing) is borne out by these non-linear data. . . Line 538/Fig 6 – as for Fig 2; let's see the raw data. Line 544/Fig 7 – these look quite non-linear; comment? Figure 7 seems an odd choice of final figure; perhaps add a sketch/cartoon of pore space, pore bodies + pore throats, and their distributions; mapped to the 'best' tools for quantifying them.

Dave Healy Aberdeen June 2018

---

## Editor Comment (EC1) · M. J. Heap (Editor) · 25 Jun 2018

Dear Mr. Zhang,

Firstly, thanks for submitting your work to Solid Earth. As you can see, I've now received three reviews of your paper. All three reviews highlight an interest in your data, but all three state that major revisions are required before your manuscript can be seen as appropriate for publication in Solid Earth. If you are willing, please now prepare a detailed point-by-point rebuttal letter and a revised manuscript.

Thanks,

Mike Heap (Topical Editor of Solid Earth)

---

## Author Comment (AC1) · 18 Jul 2018

**Dear editor, dear reviewers,**

We thank the reviewers for the careful inspection of our manuscript and providing useful comments and recommendations that helped to improve our paper. We made effort to address all the reviewer's comments. Beside numerous corrections and further text, we added new Figures (Fig. 1E, F), a new table (Table 3), and another nine references. You find our answers in blue color. We highlighted the changes in the text by yellow background.

Anonymous Referee #1

General comments:

The analysis of pore space geometry in rocks is of great relevance to many areas of earth sciences. In this study, the authors propose to compare the results obtained from different pore space characterization techniques in two sandstones, the Bentheimer and the Rotthbacher.

Though I think this is a great topic to investigate, I find that the work itself does not bring significant value in its present form. A lot of research has already been conducted in this domain and it is difficult to see what new element the paper provides, aside perhaps from the spectral induced polarization part which I unfortunately had a very hard time following. Maybe the work could be augmented with a better review of previous findings and a more thorough extraction of information from the microCT images. In the abstract, the authors announce that they are going to characterize the pore space geometry, although nothing is done beyond acquiring and tracing various cumulative curves. In the abstract again, the concept of fractals is used but it is unclear (1) whether it is warranted to compute a fractal number from these curves as it is not demonstrated that they represent distributions of objects and (2) what the authors recommend what one should do with that value.

We thank for the critical remarks. We agree that a lot of research has been done in the domain of pore space characterization regarding µCT, MICP, and NMR. The main aim of our paper is to integrate an electrical method in this study. The spectral induced polarization (SIP) method provides a relaxation time distribution. It is assumed that the relaxation times are related to geometric length in the pore space. In order to verify this approach, a comparison with other methods providing insight into the distribution of pore sizes is necessary. The curves representing the cumulative volume as a function of pore radius provide the chance to compare the curves resulting from different methods. Additionally, the cumulative curves are used to assess the fractal behavior of the pores volume distribution. The fractal dimension is a

useful number for up-and downscaling of geometrical quantities. Additionally, the fractal dimension is used in methods of permeability prediction.

Regarding the work that is done on images and the comparison between different data sets, I wish the authors had provided more information and figures on the image segmentation result as well as the result of the maximum inscribed sphere (MIS) computation. Also, the authors may be aware that such computation (MIS) can be used as a starting point for performing a digital equivalent to MIP which becomes then valuable to compare with the experimental mercury injection curve. In fact, a comparison between experimental MIP and digital MIP on one hand, and between MIS and NMR on the other would have made more sense.

Because the reader does not have access to the state of the images prior to the tracing of the cumulative 'pore size' curve (MIS), it is very difficult to check whether the result is consistent. I have a doubt regarding the offsets observed in Figures 2 and 6 and I am wondering if what is plotted for the MIS is really the radius or rather a diameter. Please check. Also I am surprised to see that virtually no 'objects' with dimension smaller than 20 microns was detected in either sandstone considering the image resolution of 1.75 microns per voxel and 1.5 microns per voxel for the Bentheimer and the Rothbacher, respectively.

As shown within figures 2 and 6, in fact pores in range of the image resolution have been detected and segmented. Nevertheless, these pores with small radii almost provide no significantly volume contribution concerning the results of the digital image analysis. We assume that this has a couple of reasons: first, the pore segmentation is greatly influenced by the density contrast between each individual phase (void and solid minerals), as well as by typical X-ray artifacts such as partial volume effects, which cannot be quantified in detail with conventional µ-CT. Additionally, the watershed algorithm will be sensitive towards the previously mentioned reasons and hence will lead to under-estimated volumes especially for small pores. We have addressed these effects within the revised manuscript.

Our study shows how other methods can be used to extend the resolution to smaller pores. We confirm that the pore radii are presented in Figures 2 and 6. The MIS data contains over 10000 points and it is a huge table so we only show the figures here. The data points of several voxels were eliminated because these data may either be pores or noise. We added a new Table 3 that compiles the minima, maxima and mean values of different pore sizes derived from µ-CT data of the two samples.

I don't think that I can speak at length to the SIP part because I am not familiar with it. I would like to see a more intuitive explanation as to why it is appropriate to compare SIP data with a drainage (MIP) curve. Also I don't understand how the data of Figure 5 on frequency-dependent complex conductivity is converted into relaxation times (assuming this is what is being done).

We recognize that SIP is a novel method in the field of pore size distribution. We added a paragraph to explain the Debye decomposition approach that is used to transform the spectra of complex conductivity into a relaxation time distribution.

In terms the organization of the paper, I found the figures confusing in the fact that they convey more information than is being discussed at first, forcing the authors to go back and forth when describing their results.

We have reconstructed the Results and the Discussion.

I am convinced the authors have at they disposal a great starting point to a valuable study. The sandstones picked are definitely materials of interest to the community and the high resolution microCT images can certainly be exploited further.

We recognize that the reviewer is most interested in the μCT images. It would have been possible to write an additional paper with special focus on the data processing and interpretation of the μCT data of the investigated samples. But the reviewer should consider that the μCT yields information, which is limited by the above described effects. The other methods are quite important to get insight into the pore space at the sub-micrometer scale. We investigated in our study how NMR, MIP, and SIP data can be used to get insight into the structure of pore size over a wide range of pore radii.

Other comments:

*I think the English could probably be improved (grammar and choice of words mostly)

We made some corrections highlighted in yellow color in the text.

*In the conclusion, an image resolution of 3.5 microns per voxel is quoted – please decide.

Correction done.

*I did not see the benefit of plotting the curves starting from the smallest injection radii - it puts an emphasis on the fraction where there is less data and also that likely contributes nearly nothing to flow, while dwarfing most of the important information.

We plotted the curves starting from the smallest resolved radii to show the different pore ranges detected from the different methods. The fractal nature of pore size can only be recognized if a wide range of pore radii is resolved.

*How were the injection steps chosen for the MIP curves? It seems nearly random, and also very sparse in the case of the Rothbacher.

The injection steps were automatically recorded by the equipment. In Figures 2 and 6 the MIP curves were plotted with step values of 10 (only every tenth value is shown) and 4 (only every forth value is shown), respectively, in order to show the data points clearly.

*Table 2 would be easier to look at if some mineral names were added to it.

We changed the term mineral phase to chemical components, and we add a reference describing the minerals of the Bentheimer sandstone.

*The resolution of the microCT images is great, the authors should be able to show much more detail at the grain scale. Have the authors attempted to determine whether the voxel dimension was a true image resolution?

Thank you for the positive evaluation of the µCT images of the two investigated sandstone samples. Certainly, the image quality is good down to a resolution of roughly two voxels. But this resolution is not sufficient to characterize the small pore space of sandstone underneath a distinct radii threshold as previously described.

D. Healy (Referee)

d.healy@abdn.ac.uk

General comments

This ms is a comparison of 4 methods for characterising pore space, especially pore size distributions. Data from tests on 2 sandstones are compared. Overall, it is a good idea, quite well presented, and should provide a useful addition to the literature. It's easy to suggest extra tests or analyses but the 4-way comparison stands as is (but see below), and the separation

into methods better suited for pore bodies and pore throats is good. I think this paper could be acceptable, subject to moderate revision. My main issues are with the relative lack of quantitative comparison of the methods, and the apparent underlying assumption of power law behaviour. E.g. in Figures 3 and 7, assuming you can make a case for power law behaviour, what is the departure of each line/method from a modelled power law prediction? The paper does not go anywhere near far enough in this regard, in my opinion.

We thank for the positive evaluation of the general idea of our manuscript with the focus on a comparison of different methods regarding their potential for a detailed consideration of pore size distributions over a wide range of radii.

Specific comments

Line 88-90 – much more clarity needed here; a log-log plot of these variables MAY show a straight line, which COULD then be interpreted as power law behaviour. Are you assuming a power law, and therefore fractal behaviour of the data? Perhaps a Maximum Likelihood Estimator approach might be relevant here (Clauset et al., 2009 SIAM Review; Rizzo et al., 2017 Journal of Structural Geology).

The reviewer is correct. The presentation of the cumulative pore volume as a function of pore radius as log-log-plot, which corresponds a power law behavior, is necessary to recognize the fractal behavior of the pore volume. A deviation from an idealized straight line indicates non-fractal behavior. We add the content of Maximum Likelihood Estimator considering the underlying assumption for power law behavior.

Line 106-107 – but this limit is rock/ CT scanner/segmentation dependent, right? So add the caveat, 'for this study, the CT resolution limit is . . .'.

Correction done.

Line 165 – 'arguable'

Correction done.

Line 199 –'extent'

Correction done.

Line 231, 243 – 'mainly'; can you be more specific about the modal proportions of minerals?
Correction done.

Line 240, 256 – what method for the porosity estimate?

Correction done.

Line 249 –'depositional'

Correction done.

Line 251 – 'shows'

Correction done.

Line 384 – 'extent'; 'differs'

Correction done.

Line 401 – 'of' not 'on'

Correction done.

Line 508 – Table 2; these are not mineral phases, they are chemical components!

Correction done.

Line 512/Fig 1 – SEM; what detector, BSE? Say so.

The detector is BSE detector.

Line 520/Fig 2 – I think we need to see 'raw' data for these methods; AND then the 'processed' data using the 'known' porosity. Let the reader judge the data.

Line 538/Fig 6 – as for Fig 2; let's see the raw data.

Figure 2 and Figure 6 are the 'raw' data, and Figure 3 and Figure 7 are the 'processed' data using the 'known' porosity.

Line 523/Fig 3 – my point above about assuming power law/fractal behaviour (NB, not the same thing) is borne out by these non-linear data. . .

Line 544/Fig 7 – these look quite non-linear; comment?

The whole curves in Figure 3 and 7 are non-linear, that is correct, so we only use an interval with a constant slope. We add the text in the Discussion: "The whole curves of the four methods are non-linear, and a Maximum Likelihood Estimator (MLE) approach might be

relevant here (Rizzo et al., 2017). For example, in the case of the NMR curve of Bentheimer sandstone, the fitting of all data using the MLE reveals that the log-normal distribution is the most likely distribution with the estimated parameters $\mu$=3.43 μm and $\sigma$=0.82 μm. These two scaling parameters are the logarithmic mean and logarithmic standard deviation, respectively".

Figure 7 seems an odd choice of final figure; perhaps add a sketch/cartoon of pore space, pore bodies + pore throats, and their distributions; mapped to the 'best' tools for quantifying them.

A. Bubeck (Referee)

Reviewers' Comments:

The submitted manuscript describes a study that aims to characterize pore size distributions, and the scaling of attributes, for two well-constrained types of sandstone using the following methods: 1) micro CT; 2) mercury intrusion porosimetry; 3) nuclear magnetic resonance; and 4) spectral induced polarization. Using curves of the cumulative volume fraction of pores and pore body radius, the methods are compared.

Generally speaking, the aims of the paper are clear, and data are well-presented. Improving our understanding of the nature of pore space in natural rocks is important to a broad range of study areas, including petrophysics and rock mechanics, for which a comparison of methodologies will be potentially very useful. It is my view, however, that the manuscript, in its present form, contributes little to our understanding of natural pore variability; resolution cut-offs for these techniques have been discussed previously. Many of the sections need rewriting to avoid overlap and repetition, and the motivation, and context for the study are unclear. For these reasons, it has been a challenge to review as thoroughly as I would like. Some simple restructuring and clarification could improve this but additional work is also necessary. For instance, the authors state early on that they provide a "multiple length scale characterization of pore geometry" - where is this data? A sentence in the conclusion section refers to "μ-CT enables a geometrical description of individual pore space. . .", but no data are presented. Providing additional geometric data would greatly improve the novelty of the work, and appeal to a broader readership. If the authors lack this data, I recommend that they remove references to geometry throughout. Furthermore, there is virtually no review of existing geological (or other) data derived using the techniques, nor their application, and a

scattered discussion of the results towards the end of the manuscript. This needs work before it can be considered for publication.

We add a table (Table 3) with the geometric data derived from μ-CT in the Results.

The authors may decide that additional data, and/or background and discussion is beyond the scope of this work. If this is the case, then considering the broad readership of Solid Earth, they might consider submitting the work to a more subject-specific journal.

We selected "Solid Earth" because the pore space distribution and geometry is of interest for a broad readership of geoscientist: geologists, petrologists, geophysicists, petrophysicists. A comparison of different methods used in this field should be a motivation not only to apply the traditional standard methods. SIP is a novel method in this field. It is our aim to check the potential of this method in pore size characterization.

As a final note, the manuscript should be given thorough proof-read from a native English speaker; language, grammar, and sentence structure require work before I think it is ready to publish. I have not made specific edits related to these because they are numerous. Below, I have provided section-specific comments for the authors.

I hope the authors find them helpful.

We thank for the critical and helpful remarks. We tried to address the remarks and added additional information as recommended by the reviewer.

MS Specific Comments

0. Abstract The abstract reads fine and, with some minor edits, reflects the main findings of the manuscript.

Line(s) Comment 39 ". . .pore space geometry. . ." Global edit: you do not discuss pore geometry in this work. Please remove reference to it unless you have additional data to add.

We add the text defining the geometry here.

1. Introduction This section needs some restructuring, and expansion. For submission to Solid Earth, the work should mention the existing applications of these methods (geological, and others). The authors could consider including a background section to present existing uses of each method, their limitations, and how this data is used. This would improve the framing of the work, and broaden its appeal to different readers.

We add some references of the applications of these methods.

These are some questions I have from this section: a. What are your motives for the study? b. Who are you targeting with this work? Petrophysicists? Is Solid Earth the best place to present this? Porosity, and its effect on fluid flow and the mechanical behaviour of porous rocks is important to a range of study areas, and the subject of extensive study. A (brief) review of some the existing literature would help - e.g. recent uses of x-ray CT to analyse pore volume/geometry/distribution and fluid flow through porous media.

Line(s)

52-57 This isn't much of an introduction for the study.

We add the text defining the geometry mentioned in this paper.

63-67 I think you could move this to your discussion section and expand on how results from these studies compare to your own.

68-75 This section isn't very clear. You mention 3 separate published NMR studies but no others? It is not immediately clear what the relevance of these studies are to your own motivation, or results. You should add some background for the other methods too. Speaking for micro CT, there are several studies that analyse rock pores that the authors should take a look at, including: Lindquist et al., 2000; Ketcham, 2005; Nakashima and Kamiya, 2007; Takahashi el al., 2016; Schmitt et al., 2016; Saenger et al., 2016; Bubeck et al., 2017; Zhao et al., 2017 (this last one should be particularly helpful in helping you structure and present your work).

We reconstructed this section.

76-83 This section is partly repeated. Read through your introduction and keep it concise.

Correction done.

77 Either: add a section that deals with pore geometry specifically, and include additional figures/data, or remove this.

Correction done.

80 ". . .distributions are connected to each other. . ." This is unclear.

Correction done.

2. Methodology

This section is very long and it looks like a mixture of background and methodology, which could be split up. Keep your methodology simple: what did you do, and to what?

A number of questions need to be addressed in this section to improve the clarity of your results. a. How many samples did you use in total? b. How many samples did you run for each test? This needs explanation to demonstrate that the results are repeatable. You could add this data to table 1. You should also explain what size samples need to be. c. Does sample size effect the results for any of the methods? It should certainly be considered in micro CT. d. Did you characterize grain size? What role will this play?

We reconstruct this section and add the sample information in Samples and methods section.

Line(s)

91-99 This is background

Correction done.

99 The use of fractals (in geology alone) stretches considerably beyond these references. If you are presenting data on pore scaling relationships, expand this and move to a background section.

We change the term 'geometric objects' to 'pores in sandstones and carbonates'.

100-102 Good. You should move your sample descriptions into this section.

Correction done.

103-108 Tell me what you're doing/using before you tell me the limitations of it.

Correction done.

109-110 How have you performed this analysis, what were the results, and why is it best? More information and figures to explain this process are needed. Why compare with 2D SEM images?

We used a standard analysis of the μ-CT data. A comparison between 2D SEM and a 2D slice of the CT image is shown in Figure 1. We add a Figure showing 3D images of pore network resulting from μ-CT.

113 ". . . each individual pore. . ." Where are your data for individual pores? You could present data on the proportion of pore body radii (how variable are they within samples? How do your different sandstone types compare?), their geometry and preferred orientation (if present).

We add a new Table 3 summarizing the geometric information resulting from µ-CT. The data of individual pores contains over 10000 data points so the table is not shown in the paper.

114 Do you have references for this technique?

Correction done.

119 You're using MIP specifically for pore throat radius? Be clear about what data each method is providing!

131-132 Try to explain this more clearly.

The detailed explanation can be found in the reference. You can imagine the big pore as a room with a door (throat), and the MIP data records the size of the door instead of the room.

139-145 Background.

Correction done.

142 "a capillary model with cylindrical pores, of uniform radius. . ." - How useful is this? Is it reasonable to assume a linear relationship for natural pores, which may be neither cylindrical, nor uniform? Kleinberg, 1996 applies it to "slit-like pores". You should provide some geometric description of your pore space to support the use of this model. At the VERY least, explain how your results could vary. You could leave this as is here, but add a section to your discussion covering possible limitations of the method and expand on this there.

The reviewer addresses an interesting issue. The surface to volume ratio is not very sensitive to the shape of the pores. As mentioned in the text, a cylindrical pore results in a surface to volume ratio of 2/r and a spherical pore in 3/r. The resulting difference (factor 1.5) can be ignored when looking at a logarithmic scale. The word "uniform" is deleted. A cylindrical pore has a constant (uniform) radius per definition. The geometric descriptors are compiled in Table 3.

150-151 "The range of resolved pore radii depends on the used value of surface reflexivity." – You need to explain what value you have chosen, and why? If similar assumptions apply to the use of a single value here, address it in the discussion also.

The correct choice of surface relaxivity is not a simple issue. We have demonstrated in chapter 4 a procedure to determine $\rho$ from a comparison of μCT and NMR. This procedure demonstrates a joint use of different methods improves the reliability of the derived parameters. Applying this procedure, we got different surface relaxivities for the Bentheimer ($\rho$ = 54 μm/s) and Röttbacher sandstone ($\rho$ = 237 μm/s).

175 Quantify the "restricted range"

Correction done.

182 Did you have any repeatability issues? Do you think two sufficient?

In most cases, two measurements are sufficient for checking the repeatability of the acquired complex conductivity spectra. If the two measured spectra show remarkable differences, additional measurements are performed until no temporal changes are performed.

187-224 This section is a mixture of background, discussion and results, with some repetition. Read through carefully and move elsewhere where appropriate.

Correction done.

3. Sample material This section is a mixture of methodology and results. Does it need to be a separate section? I think it would be helpful to have this information earlier. When you're describing the methods, it will help to know what they are related to. The section could also be much shorter. I recommend you edit the information into your methodology, explaining the number of samples used and their sample sizes

Correction done.

Line(s)

227-228 Cut this. Unnecessary.

Correction done.

244-245 Remove 245-247 Background

We think that this additional information might be useful for the reader familiar with different types of sandstones.

251-262 This is your data? It needs to be in the results section.

Correction done.

4. Results The description of the graphs in this section is rather vague and unhelpful - you refer to "differences" between curves but don't describe or quantify these. Equally, stating that something is "larger" or "smaller" is also unhelpful. Quantify your observations!

I suggest changing the section titles here to refer to the parameters measured, i.e. pore volume fraction, pore throat radius, and scaling. Describe the results of these for each rock sample. This would be easier to follow than describing each rock; it is the technique that should be the focus of this section.

Correction done.

Your fractal data is currently lost in your discussion section. Move it into your results and consider a table that compares the dimensions obtained.

The reviewer finds all the derived parameters of our study in Table 1 including the fractal dimensions for the two samples.

Line(s)

282 ". . .wide range of pore radii. . ." State the range measured for each method, with

upper and lower limits clearly defined.

Correction done.

289 Explain this more clearly.

We describe in line 289 the observation in Figure 2 that the μ-CT curve is shifted to larger pore radii in comparison with MIP.

5. Discussion

This section is a confusing mixture of results and discussion. You should read through it carefully and remove results to earlier sections.

After reading the discussion, I have a number of questions that can be addressed by restructuring and expanding the discussion. a. What are the implications for the pore size distributions you have obtained? There is no real discussion of the importance of pore size distributions, or their use, in this work. . . b. What effect do you think your results have for studies of the mechanical behaviour and fluid flow properties of your samples? c. How do your results compare to existing characterisations of these samples - are you offering an improved resolution? How important is it for samples like this? d. How strongly dependent are your resolutions on rock type? How would they vary for other rock types: Limestones/volcanic rocks for example? e. Are the approaches described suitable for a range of rock types/sample sizes/porosities? I would like to see this section discuss the implications of the results more broadly. f. Which of the techniques is best? Which one provides the most, useful, information -and for whom? g. What are the limitations of your findings?

We reconstruct the Discussion section.

A summary table of the data provided by each method, and resolution would be helpful to readers.

We guess that the reviewer has overlooked Table 1, which is summarizing the parameters derived for the two samples of our study.

Also include a limitations section and use this as a basis to explain which methods should be used for certain applications, and how cautious researchers should we be when interpreting the results.

6. Conclusions A lot of the material in this section could be moved into the discussion.

Add a sentence summarising the importance of the result of this study.

Line(s) 399-407 These are (some of) your conclusions. The rest can be moved.

7. Figures/Tables

Suggested new figures: a. Demonstrate the 3D pore network would be helpful when describing your samples (e.g. Figure 3 in Zhao et al., 2017)

b. Segmentation/MIS

We added 3D images (Figure 1 E, F) showing the pore network for the two samples.

c.Plot your body radii against cumulative frequency.

We do not see the purpose of such a plot. We prefer to plot the cumulative volume as a function of pore radii as explained in the text, because this graph in a log-log-plot enables the determination of the fractal dimension.

Figure 1 No changes suggested.

Figure 2 Label the two values (0.238 and 0.184) in the caption.

Correction done.

Figure 3 No changes suggested.

Figure 4 Label each part as A and B; describe them separately in the caption.

The curves of the two samples are displayed with different symbols and can be easily differentiated.

Figure 5 No changes suggested.

Figure 6 Label values as for figure 2.

Correction done.

Figure 7 No changes suggested.

Table 1 Where does your permeability data come from? I recommend you convert these values to m2.

The gas permeability has been measured and Klinkenberg correction has been applied. The unit mD has been converted into m².

Table 2 No changes suggested.

---

## Author Response (AR2)

**Dear editor, dear reviewers,**

We thank the editor and reviewers for the critical comments to our paper. We made effort to address all comments. You find our answers in blue color. We highlighted the changes in the text by yellow background.

Topical Editor Decision: Reconsider after major revisions (27 Aug 2018) by Michael Heap

Comments to the Author:

Dear Mr. Zhang,

Based on the concerns of the first set of reviews, I sent your manuscript for re-review. I now have two new review reports: one that recommends "rejection" and another that recommends "major revision". Their concerns centre around three issues: (1) missing references to key works, (2) insufficient discussion of the data and, importantly, (3) the absence of significant novelty. Because I think that these concerns can be remedied, I'd like to give you the opportunity to further improve your manuscript and resubmit to Solid Earth. I have therefore selected "major revisions". However, please be aware that I consider that addressing these comments will require large changes to the manuscript, rather than the addition of a few new sentences. If you are willing, please now prepare and upload a detailed point-by-point rebuttal letter and a revised manuscript.

Thanks,

Mike Heap (Topical Editor of Solid Earth)

We thank the Topical Editor of Solid Earth to provide us a further opportunity to improve our manuscript.

We have done much effort in addressing the comments and suggestions of the three reviewers in the first iteration of the reviewing process. Beside numerous corrections and further text, we added new Figures (Fig. 1E, F), a new table (Table 3), and another nine references. Another recommendation of a "major revision" indicates that the improvements of the manuscript were neither fully acknowledged by the editor nor by the two reviewers of the second iteration of the review process.

Unfortunately, the remarks and recommendations of the two reviewers are not quite helpful for a further improvement of our manuscript. A general evaluation of the presented approach

is missing. The critical remarks of the two reviewers are related to minor issues of special methods and do not consider the integral inspection of data resulting from different methods with varying resolution, which is the main focus of our study.

Reviewer #1 recommends a rejection of the manuscript because he/she does not recognize the novelty of our approach and claims that the used Debye decomposition should be replaced by Warburg decomposition. We do not agree with the argumentation of this reviewer and address these issues in our "rebuttal letter" below.

Reviewer #2, who has already reviewed our manuscript in the first iteration, recommends further clarification of some issues that have already been explained in the text.

Nevertheless, we follow the recommendation of the Topical Editor and

(1) added eight more references to key works;

(2) improved the discussion of the data;

(3) indicated the difference of our approach in comparison to previous studies.

We hope that our additions will clarify the key messages of our study.

Report #1

The paper presents experimental data of different nature including NMR and SIP in order to characterize the pore size distribution of sandstones. In my opinion, there is nothing fundamentally new with respect to the work done by Niu Q. and Zhang C. 2017. Joint inversion of NMR and SIP data to estimate pore size distribution of geomaterials. Geophysical Journal International.

We are aware of this interesting paper of Niu & Zhang (2017). The reviewer should recognize that our manuscript does not propose a "joint inversion" of NMR and SIP data to get a more reliable pore size distribution. The paper of Niu & Zhang (2017) assumes that NMR and SIP resolve the same pore geometry. We and other authors (e.g. Revil et al., 2014) assume that SIP (and MIP) resolve the pore throat and μ-CT and NMR the pore body radius. Considering the different geometric parameter, a joint inversion can only be done if the ratio between pore body and pore throat radius it known. We spent effort in our study to get this ratio from the comparison of MIP and μ-CT data.

The second biggest problem I see in this paper is the use of the Debye deocmpostion which is fundamentally in error with respect to the basic physics of the problem. If my understanding is correct, it was shown that such a kernel inplies that if all the pore have the same size, the spectra are described by a Debye model. No metallic-free rocks is described by a Debyt. At best (very uniform materials), the transfer function is a Warburg (a Cole-Cole with an exponent c = ½ to be compared with a Debye for which c = 1). Using a Debye decomposition is therefore a huge mistake in recovering the pore size distribution. Why this has been described in many papers in the last few years since Florsch N., et al, Inversion of generalized relaxation time distributions with optimized damping parameter, Journal of Applied Geophysics, 109, 119–132, 2014., it seems that the authors are not aware of these works. I think they should spend more time in reviewing precisely the literature on the subject and redo the analysis with a Warburg decomposition instead of a Debye decomposition.

We thank the reviewer for addressing this issue, but we do not agree with this opinion. We know the interesting paper of Florsch et al. (2014) and acknowledge their work to generalize the decomposition of IP spectra using different kernels, but you will not find any statement in this paper that a Warburg decomposition is the only procedure to get a "true" pore or grain size distribution. This paper describes the mathematical and numerical procedures to get distributions based on different models.

There is a variety of papers that use the idea to overlay Debye models with different relaxation times to simulate IP spectra of media with a wider pore (or grain size) distributions (e.g. Leroy et al., 2008; Nordsiek & Weller, 2008; Revil & Florsch, 2010; Niu & Zhang, 2017). Please, have a closer look at the recent paper of Niu & Zhang (2017) which you mentioned in your review. Their approach is based on a Debye kernel as well (see their equations 5, 8, 12, and 13). I guess that no reviewer has recommended to reject this paper because of the used superposition of Debye models.

The paper of Revil et al. (2014), which has already been referenced in our manuscript, compares Debye (DD) and Warburg composition (WD) for a set of six sand samples. Regarding the peak and width of the resulting pore size distributions (but not the location), they show evidence that WD of IP spectra results in (a slightly) better agreement with the pore throat distribution from MIP. Having a closer look at their Figures 17 to 19, which indicate a rather good similarity between DD and WD results, this investigation needs "additional data … to confirm this finding" as stated by the authors. We recognize that for two of the six samples (436 and 499) a better agreement of the peaks of DD and MIP pore

size distributions. Therefore, we cannot recognize that the use of DD is a "huge mistake" that contradicts the physical behavior as stated by the reviewer.

Certainly, more studies have to be done to check different approaches. Our results, which are based on DD, demonstrate a fairly good agreement between the pore size distribution derived from SIP and MIP. A careful comparison between DD and WD (based on quantitative criteria and a large set of samples) is outside the main scope of this paper and cannot be done in three weeks given for the revision.

Other issues include a very poor review of the existing literature on these techniques in the introduction.

Considering the remarks of the reviewer, we extended the review of existing literature. We added more references in the Theory section, where the methods are described.

I also disagree with this statement "Rouquerol et al. (1994) reported that no experimental method provides the absolute value of parameters such as porosity, pore size, surface area".

Unfortunately, the reviewer does not provide any justification for the disagreement. In order to avoid misunderstanding, we extent the text after this reference with remark to the fractal nature of the mentioned parameters.

I am also surprised that the earlier works by Slater and Lesmes are not cited.

One of the co-authors has got a successful collaboration with Lee Slater for many years and is aware of his earlier works with David Lesmes. The two authors published an interesting paper (Slater & Lesmes, 2002) relating the grain size ($d_{10}$) to the imaginary part of conductivity. The reviewer is correct that this paper may be regarded as an early contribution to combine geometric and IP parameters, but the idea to relate pore size distributions to relaxation time distributions was proposed later. We added this reference with others to acknowledge the early contribution.

No discussion is given for the small pore sizes that are hidden by the Maxwell Wagner relaxation.

The reviewer is correct that we did not mention explicitly the Maxwell Wagner polarization. We wrote in the text (lines 216 – 218):

"Considering that the complex conductivity spectra are affected by electromagnetic coupling effects or other polarization effects at higher frequencies and by a lower signal to noise ratio for lower frequencies, we focus on the frequency range between 0.01 Hz and 100 Hz." We replace "other polarization" by "Maxwell Wagner polarization and dielectric effects". We add the following sentence:

"Smaller pore sizes are hidden by Maxwell Wagner relaxation and dielectric effects that are not easily related to pore geometry."

Report #2

Though I appreciate the efforts made by the authors to respond to my earlier comments, I still feel that this contribution is lacking essential material to support the results that are reported.

We have spent much effort to address the comments of all reviewers in our first revision. We have to apologize that we were not able to follow all recommendations of the reviewers. We paid attention to support the key messages of our study. We are aware that additional material can be included to clarify problems encountered by the different methods. But, we would like to present a concise manuscript that focuses on the key messages. A paper that combines the results of four different methods is not the place to provide and discuss details of each individual method. We preferred to provide a variety of references to the theory of the individual methods.

The assimilation of the MIP data to a pore throat size distribution on one hand and of the NMR data to a pore body size distribution is not substantiated.

A careful look into the text shows that we have provided an explanation for the different pore sizes resulting from MIP and NMR. We have described in the text of chapter 3 that mercury intrudes into the pores through the pore throats:

"Starting with low pressure, the pores with larger pore throats are filled with mercury. While increasing the pressure, the pores with smaller throats are filled. Reaching a certain pressure level $P_c$, a cumulative volume of mercury ($V_{Hg}$) has intruded into the sample that corresponds to the pore volume being accessible by pore throats radii larger or equal $r_t$ according to Eq. (2)."

We have shifted this explanation into the Theory chapter and have added a remark to the new Figure 1, which shows a 2-D image of the pore space with pore throats and pore bodies.

We have added a remark in the Theory chapter related to NMR:

"It should be noted that the NMR method resolves the radius $r_b$ that corresponds to the maximal distance to the pore wall. It can be represented by the pore radius of the largest sphere that can be placed inside this pore as shown in Figure 1."

The authors do not offer a single image of a pore space representing what they think the techniques are measuring.

Following the recommendation of the reviewer, we have added the new Figure 1, which shows a 2-D image resulting from μ-CT of the pore space with pore throats and pore bodies. Using this Figure, we explain which radius is measured by the different methods.

There is more than one way to generate distributions such as the ones showed for the pore body radius, and the very large contrast that is observed with the MIP results should warrant further investigation, starting with reviewing graphically how pore bodies are segmented and tagged. Then one would have to explain what these individual pores might have to do with the NMR signal.

We display the distribution of the pore body radius resulting from two different methods: NMR and μ-CT. The algorithm providing the pore body radius from NMR $T_2$ relaxation times is based on equation 5 as described in chapter 2. The uncertainty of the position of the NMR curve (in horizontal direction) is related to the unknown parameter of surface relaxivity. According to our experience, we determined the pore radius from μ-CT images using the largest sphere that can be placed inside each individual pore (maximum inscribed sphere method, e.g. Silin and Patzek, 2006, see new Figure 1). We agree that another approach would possibly result in a slightly different distribution.

As already mentioned in the previous review, I also think that the data set could be made more complete by adding an MIP simulation based on the images as well as a distribution of the maximum inscribed sphere radii in the pore space prior to pore separation.

We agree that it is possible to perform a MIP simulation based on the 3-D µ-CT data. Results can be found amongst others in Knackstedt et al. (1998). Considering the resolution of µ-CT (in this study: no pores considered with radii < 10 µm), most of the pore throats (< 10 µm) are not resolved. Therefore, an agreement between "synthetic" MIP data and measured MIP cannot be expected. A reliable MIP simulation requires sufficient resolution of both pore bodies and pore throats. In the case of our samples, we find that µ-CT resolves the largest pores. This is a strict limitation of the µ-CT method.

According to our approach, the cumulative volume distribution resulting from µ-CT can be "continued" to smaller pore radii after an adjustment of the NMR curve by selecting a suitable value for surface relaxivity. From a technical point of view, i.e. due to the DIA software used, pore separation is essential before performing the maximum inscribed sphere algorithm. Otherwise the "entire pore volume" will be approximated by one equivalent sphere. Please be aware, that the "in-situ" image is used for analysis, we do not perform pore network modeling in order to derive equivalent pore body diameters.

Reference:

Knackstedt, M., Shepard, A.P., and Pinczewski, W.V. (1998): Simulation of mercury porosimetry on correlated grids: Evidence for extended correlated heterogeneity at the pore scale in rocks. Phys. Rev. E, 58, R6923(R), 1998.

---

## Author Response (AR3)

Authors comments on the topical editor's comments and suggestions

Dear Mr. Heap,

Thank you for your comments and suggestions concerning our manuscript. We submit an updated version of our manuscript that considers most of your suggestions. We highlighted the changes in the text by yellow background.

Although reviewer #1 disagrees with the use of the Debye model, I appreciate that the authors now include a more detailed reasoning. However, I would like the authors to include more information that outlines why Revil et al. (2014) consider that, to quote their paper: "…the Warburg function, rather than the Debye function, is the correct transfer function to determine the pore size or the pore size distribution". Revil et al. (2014) provide a lengthy discussion on this topic. Since the authors disagree with the reasoning of Revil et al. (2014), I would like them, in a few sentences or short paragraph, to explain why they disagree and why they have decided to use the Debye model.

We have extended the paragraph in the Theory chapter to explain our motivation to use the Debye decomposition:

*"Florsch et al. (2014) demonstrated that a variety of models can be used as kernel for the decomposition of the spectra. Revil et al. (2014) compare the results of Debye and Warburg decomposition. Their argumentation, which is based on mechanistic grain size models describing the polarization of charged colloidal particles and granular material, supports the application of the Warburg decomposition that results in a narrower distribution of polarization length scales. It should be noted that a uniform grain size does not automatically generate a uniform pore size. Besides it can be clearly seen by the scanning electron microscopy images, that the investigated sandstones feature a distinct range of both, grain and pore (throat) sizes. Considering that the pore size and not the grain size controls the polarization of sandstones, as observed and by different authors (e.g. Scot & Barker, 2003; Niu & Revil, 2016), a wider distribution of length scales can be expected. According to our opinion, there are no clear indications for superiority of the Warburg decomposition. Up to now, a theoretical model that confirms the validity of the Warburg model in describing the polarization of a simple pore space geometry has not been presented. Therefore, we prefer to use the Debye decomposition, which has proved to be a useful tool in the processing of IP data in both time and frequency domain (e.g. Terasov and Titov, 2007; Weigand and Kemna, 2016)."*

There are still several instances where the authors mention that MIP yields the pore radii, rather than the pore throat radii. For example, on line 373: "The MIP yields the widest range of pore radii." Due to the subject matter of the paper, I think it's important to be precise here and elsewhere in the manuscript.

We agree with this point and have been more precise within distinct parts of the manuscript using "pore radii", "pore body radii" and "pore throat radii", respectively.

I think that the authors should offer more information on their method of permeability. Gas permeability measurements on porous sandstones typically require a Forchheimer correction. Did you correct for turbulent flow? Under what confining pressure were the measurements performed?

We have added additional information on the permeability measurements at the end of the Methods section of the manuscript:

*"Permeability measurements have been performed by using a steady-state gas permeameter (manufactured by Westphal Mechanik, Celle, Germany), using nitrogen as the flowing fluid. This device features a so called "Fancher-type" core holder as described by Rieckmann (1970). With this special type of core holder, significantly lower confining pressures are needed than by using a conventional "Hassler-type" core holder (12 bar for the "Fancher-type" core holder versus min. 35 - 50 bar for the "Hassler-type" core holder), leading to much less initial mechanical influence (compaction) upon the sample material. Measurements have been derived under steady-state flow conditions with accordingly low flow rates in range from 3 to 5 ml/min, leading to measured pressure drops in range from 2 to 7 mbar from sample inlet to outlet. The derived apparent permeability values have been corrected, to address the Klinkenberg-effect of gas slippage (Klinkenberg, 1941; API, 1998). Due to the usage of a steady-state technique with low gas flow rates, correction of the Forchheimer effect of inertial resistance can be neglected (API, 1998)."*

Regarding the comment of reviewer #1 about the "very poor review of the existing literature", the 2005 paper of Louis et al., for example, discusses the microstructure of Bentheim and Rothbach sandstone. Perhaps the authors can compare their results with those in this paper?

The mentioned paper in general deals with anisotropy of susceptibility and p-wave velocity. It does not give any comparable insight towards the pore structure (though it is mentioned in the headline, only grain size distributions > 50 μm equivalent diameter from 2D analysis are shown). Besides, Rothbach (Vogesen, France) and Röttbacher (Mainz, Germany), sandstones are from different locations, though they are both related to the Bunter Sandstone and hence to Triassic age. It would not be reasonable to compare different locations with each other without having all relevant data.

Nevertheless, we have added seven more references in order to improve the review of existing literature.

There's a typo on line 87: "intention".

We have corrected the spelling of this word.